# Improving and Generalizing Flow-Based Generative Models with Minibatch Optimal Transport

**Alexander Tong**[*]  *alexander.tong@mila.quebec*
*Mila – Québec AI Institute, Université de Montréal*

**Kilian Fatras**[*]  *kilian.fatras@mila.quebec*
*Mila – Québec AI Institute, McGill University*

**Nikolay Malkin**[*]  *nikolay.malkin@mila.quebec*
*Mila – Québec AI Institute, Université de Montréal*

**Guillaume Huguet**  *guillaume.huguet@mila.quebec*
*Mila – Québec AI Institute, Université de Montréal*

**Yanlei Zhang**  *yanlei.zhang@mila.quebec*
*Mila – Québec AI Institute, Université de Montréal*

**Jarrid Rector-Brooks**  *jarrid.rector-brooks@mila.quebec*
*Mila – Québec AI Institute, Université de Montréal*

**Guy Wolf**  *guy.wolf@umontreal.ca*
*Mila – Québec AI Institute, Université de Montréal*
*Canada CIFAR AI Chair*

**Yoshua Bengio**  *yoshua.bengio@mila.quebec*
*Mila – Québec AI Institute, Université de Montréal*
*CIFAR Senior Fellow*

**Reviewed on OpenReview:** *https://openreview.net/forum?id=CD9Snc73AW*

## Abstract

Continuous normalizing flows (CNFs) are an attractive generative modeling technique, but they have been held back by limitations in their simulation-based maximum likelihood training. We introduce the generalized *conditional flow matching* (CFM) technique, a family of simulation-free training objectives for CNFs. CFM features a stable regression objective like that used to train the stochastic flow in diffusion models but enjoys the efficient inference of deterministic flow models. In contrast to both diffusion models and prior CNF training algorithms, CFM does not require the source distribution to be Gaussian or require evaluation of its density. A variant of our objective is *optimal transport CFM* (OT-CFM), which creates simpler flows that are more stable to train and lead to faster inference, as evaluated in our experiments. Furthermore, we show that when the true OT plan is available, our OT-CFM method approximates dynamic OT. Training CNFs with CFM improves results on a variety of conditional and unconditional generation tasks, such as inferring single cell dynamics, unsupervised image translation, and Schrödinger bridge inference. The Python code is available at `https://github.com/atong01/conditional-flow-matching`.

---

[*]Equal contribution

# 1 Introduction

Generative modeling considers the problem of approximating and sampling from a probability distribution. Normalizing flows, which have emerged as a competitive generative modeling method, construct an invertible and efficiently differentiable mapping between a fixed (*e.g.*, standard normal) distribution and the data distribution (Rezende & Mohamed, 2015). While original normalizing flow work specified this mapping as a static composition of invertible modules, continuous normalizing flows (CNFs) express the mapping by a neural ordinary differential equation (ODE) (Chen et al., 2018). Unfortunately, CNFs have been held back by difficulties in training and scaling to large datasets (Chen et al., 2018; Grathwohl et al., 2019; Onken et al., 2021).

Meanwhile, diffusion models, which are the current state of the art on many generative modeling tasks (Dhariwal & Nichol, 2021; Austin et al., 2021; Corso et al., 2023; Watson et al., 2022b), approximate a *stochastic* differential equation (SDE) that transforms a simple density to the data distribution. Diffusion models owe their success in part to their simple regression training objective, which does not require simulating the SDE during training. Recently, (Lipman et al., 2023; Albergo & Vanden-Eijnden, 2023; Liu, 2022) showed that CNFs could also be trained using a regression of the ODE's drift similar to training of diffusion models, an objective called *flow matching* (FM). FM was shown to produce high-quality samples and stabilize CNF training, but made the assumption of a Gaussian source distribution, which was later relaxed in generalizations of FM to more general manifolds (Chen & Lipman, 2024), arbitrary sources (Pooladian et al., 2023), and couplings between source and target samples that are either part of the input data or are inferred using optimal transport. The **first main contribution** of the present paper is to propose a unifying *conditional flow matching* (CFM) framework for FM models with arbitrary transport maps, generalizing existing FM and diffusion modeling approaches (Table 1).

A major drawback of both CNF (ODE) and diffusion (SDE) models compared to other generative models (*e.g.*, variational autoencoders (Kingma & Welling, 2014), (discrete-time) normalizing flows, and generative adversarial networks (Goodfellow et al., 2014)), is that integration of the ODE or SDE requires many passes through the network to generate a high-quality sample, resulting in a long inference time. This drawback has motivated work on enforcing an optimal transport (OT) property in neural ODEs (Tong et al., 2020; Finlay et al., 2020; Onken et al., 2021; Liu, 2022; Liu et al., 2023b), yielding straighter flows that can be integrated accurately in fewer neural network evaluations. Such regularizations have not yet been studied for the full generality of models trained with FM-like objectives, and their properties with regard to solving the dynamic optimal transport problem were not empirically evaluated. Our **second main contribution** is a variant of CFM called *optimal transport conditional flow matching* (OT-CFM) that *approximates dynamic OT via CNFs*. We show that OT-CFM not only improves the efficiency of training and inference, but also leads to more accurate OT flows than existing neural OT models based on ODEs (Tong et al., 2020; Finlay et al., 2020), SDEs (De Bortoli et al., 2021; Vargas et al., 2021), or input-convex neural networks (Makkuva et al., 2020). Furthermore, an entropic variant of OT-CFM can be used to efficiently train a CNF to match the probability flow of a Schrödinger bridge. **We show that in the case where the true transport plan is sampleable, our methods approximate the dynamic OT maps and Schrödinger bridge probability flows for arbitrary source and target distributions with simulation-free training.**

In summary, our contributions are:

(1) We introduce a generalized formulation of the recent *conditional flow matching* framework (§3.1), and prove its correctness encompassing many existing flow matching methods (Lipman et al., 2023; Albergo & Vanden-Eijnden, 2023; Liu et al., 2022a) (See Table 1).

(2) We consider a special case of CFM that draws source and target samples according to an optimal transport plan, allowing us to solve the dynamic OT and Schrödinger bridge problems in a simulation-free way, using only static OT maps between marginal distributions. We show that efficient minibatch approximations to the OT map still yield correct solutions to the generative modeling problem while incurring a low detriment to the dynamic OT solution (§3.2).

(3) We evaluate CFM and OT-CFM in experiments on single-cell dynamics, image generation, unsupervised image translation, and energy-based models. We show that the OT-CFM objective leads to more efficient training and decreases inference time while finding better approximate solutions to the dynamic OT

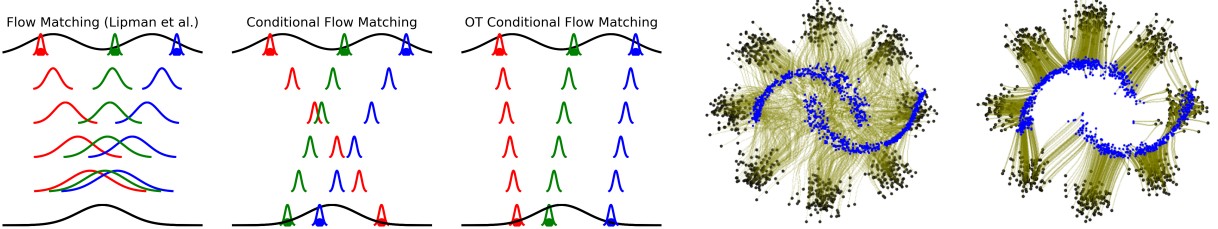

Figure 1: **Left:** Conditional flows from FM (Lipman et al., 2023), I-CFM (§3.2.2), and OT-CFM (§3.2.3). **Right:** Learned flows (green) from `moons` (blue) to `8gaussians` (black) using I-CFM (centre-right) and OT-CFM (far right).

and Schrödinger bridge problems. For high-dimensional image generation, we also propose improved and reproducible training practices for flow-based models that significantly improve the performance of algorithms from past work (§5).

(4) We release a Python package, `torchcfm`, that unifies new and existing algorithms for training flow-based generative models under a shared interface and provides implementations of our main experiments. The Python code is available at `https://github.com/atong01/conditional-flow-matching`.

## 2 Background: Optimal transport and neural ODEs

Throughout the paper, we consider the setting of a pair of data distributions over $\mathbb{R}^d$ with (possibly unknown) densities $q(x_0)$ and $q(x_1)$ (also denoted $q_0$, $q_1$). Generative modeling considers the task of fitting a mapping $f$ from $\mathbb{R}^d$ to $\mathbb{R}^d$ that transforms $q_0$ to $q_1$, that is, if $x_0$ is distributed with density $q_0$ then $f(x_0)$ is distributed with density $q_1$. This includes both the typical case when $q_0$ is an easily sampled density, such as a Gaussian, and the case when $q_0$ and $q_1$ are empirical data distributions available as finite sets of samples.

### 2.1 ODEs and probability flows

A smooth[1] time-varying vector field $u : [0, 1] \times \mathbb{R}^d \to \mathbb{R}^d$ defines an ordinary differential equation:

$$dx = u_t(x)\, dt, \tag{1}$$

where we use the notation $u_t(x)$ interchangeably with $u(t, x)$. Denote by $\phi_t(x)$ the solution of the ODE (1) with initial condition $\phi_0(x) = x$; that is, $\phi_t(x)$ is the point $x$ transported along the vector field $u$ from time 0 up to time $t$.

Given a density $p_0$ over $\mathbb{R}^d$, the integration map $\phi_t$ induces a pushforward $p_t := [\phi_t]_\#(p_0)$, which is the density of points $x \sim p_0$ transported along $u$ from time 0 to time $t$. The time-varying density $p_t$, viewed as a function $p : [0, 1] \times \mathbb{R}^d \to \mathbb{R}$, is characterized by the well-known *continuity equation*:

$$\frac{\partial p}{\partial t} = -\nabla \cdot (p_t u_t) \tag{2}$$

and the initial conditions $p_0$. Under these conditions, $u$ is said to be a *probability flow ODE* for $p$, and $p$ is the *(marginal) probability path*[2] generated by $u$.

**Approximating ODEs with neural networks.** Suppose the probability path $p_t(x)$ and the vector field $u_t(x)$ generating it are known and $p_t(x)$ can be tractably sampled. If $v_\theta(\cdot, \cdot) : [0, 1] \times \mathbb{R}^d \to \mathbb{R}^d$ is a time-dependent vector field parametrized as a neural network with weights $\theta$, $v_\theta$ can be regressed to $u$ via

---

[1]To be precise, to ensure the uniqueness of integral curves (and thus of the corresponding flow), we assume the vector field $u$ is at least locally Lipschitz in $x$ and Bochner integrable in $t$.

[2]The terminology is due to $t \mapsto p_t$ being a path on the infinite-dimensional manifold of probability distributions on $\mathbb{R}^d$.

the **flow matching (FM)** objective:

$$\mathcal{L}_{\mathrm{FM}}(\theta) := \mathbb{E}_{t\sim\mathcal{U}(0,1),x\sim p_t(x)} \|v_\theta(t,x) - u_t(x)\|^2. \tag{3}$$

Lipman et al. (2023) used a version of this objective with a stochastic regression target to fit ODEs that map a Gaussian density $q_0$ to a target $q_1$. However, this objective becomes intractable for general source and target distributions. In §3, we develop generalizations that allow more flexible and efficient generative modeling.

**The case of Gaussian marginals.** Consider the special case of an ODE whose marginal densities are Gaussian: $p_t(x) = \mathcal{N}(x \mid \mu_t, \sigma_t^2)$. While the ODE that generates these marginal densities is not unique, one of the simplest is the one that satisfies

$$\phi_t(x_0) = \mu_t + \sigma_t\left(\frac{x_0 - \mu_0}{\sigma_0}\right), \tag{4}$$

which is unique by the following theorem.

**Theorem 2.1** (Theorem 3 of Lipman et al. (2023)). *The unique vector field whose integration map satisfies (4) has the form*

$$u_t(x) = \frac{\sigma_t'}{\sigma_t}(x - \mu_t) + \mu_t', \tag{5}$$

*where $\sigma_t'$ and $\mu_t'$ denote the time derivative of $\sigma_t$ and $\mu_t$, respectively, and the vector field $u$ with initial conditions $\mathcal{N}(\mu_0, \sigma_0^2)$ generates the Gaussian probability path $p_t(x) = \mathcal{N}(x \mid \mu_t, \sigma_t^2)$.*

## 2.2 Static and dynamic optimal transport

The (static) optimal transport problem seeks a mapping from one measure to another that minimizes a displacement cost. The case of greatest interest is the 2-Wasserstein distance between distributions $q_0$ and $q_1$ on $\mathbb{R}^d$ with respect to the Euclidean distance cost $c(x,y) = \|x - y\|$. The corresponding optimization problem is

$$W(q_0, q_1)_2^2 = \inf_{\pi\in\Pi} \int_{\mathbb{R}^d\times\mathbb{R}^d} c(x,y)^2 \, d\pi(x,y), \tag{6}$$

where $\Pi$ denotes the set of all joint probability measures on $\mathbb{R}^d \times \mathbb{R}^d$ whose marginals are $q_0$ and $q_1$. For compactly supported distributions and for the ground cost $c(x,y) = \|x - y\|$, the set of solutions of (6) is not empty Villani (2009), and $W_2$ is a metric on the space of probability distributions on $\mathbb{R}^d$ with finite second moment.

The dynamic form of the 2-Wasserstein distance is defined by an optimization problem over vector fields $u_t$ that transform one measure to the other:

$$W(q_0, q_1)_2^2 = \inf_{p_t, u_t} \int_{\mathbb{R}^d} \int_0^1 p_t(x)\|u_t(x)\|^2 \, dt \, dx, \tag{7}$$

with $p_t \geq 0$ and subject to the boundary conditions $p_0 = q_0$, $p_1 = q_1$, and the continuity equation (2). The equivalence between the dynamic and static optimal transport formulations was first proven in Benamou & Brenier (2000) under the assumptions that $q_0$ and $q_1$ are compactly supported distributions with bounded density. We refer to (Ambrosio & Gigli, 2013, Chapter2) for a recent overview on optimal transport the relation between the two formulations.

Tong et al. (2020); Finlay et al. (2020) showed that CNFs with $L^2$ regularization approximate dynamic optimal transport. For general marginals, however, these models required integrating over and backpropagating through tens to hundreds of function evaluations, resulting in both numerical and efficiency issues. We aim to avoid these issues by directly regressing to the vector field in a simulation-free way.

Optimal transport is also related to the Schrödinger bridge (SB) problem (Léonard, 2014b). We show in §3.2.4 that a variant of the algorithm we propose recovers the probability flow of the solution to a SB problem with a Brownian motion reference process.

## 3 Conditional flow matching: ODEs from static couplings

### 3.1 Vector fields generating mixtures of probability paths

Suppose that the marginal probability path $p_t(x)$ is a mixture of probability paths $p_t(x|z)$ that vary with some conditioning variable $z$, that is,

$$p_t(x) = \int p_t(x|z)q(z)\,dz, \tag{8}$$

where $q(z)$ is some distribution over the conditioning variable. If the probability path $p_t(x|z)$ is generated by the vector field $u_t(x|z)$ from initial conditions $p_0(x|z)$ (see §2.1), then the vector field

$$u_t(x) := \mathbb{E}_{q(z)}\frac{u_t(x|z)p_t(x|z)}{p_t(x)} \tag{9}$$

generates the probability path $p_t(x)$, under some mild conditions:

**Theorem 3.1.** *The marginal vector field (9) generates the probability path (8) from initial conditions $p_0(x)$.*

All proofs appear in Appendix A. This result extends (Lipman et al., 2023, Theorem 1) to general conditioning variables and delineates some minor conditions on $q(z)$.

**A regression objective for mixtures.** We are interested in the case where conditional probability paths $p_t(x|z)$ and vector fields $u_t(x|z)$ are known and have a simple form, and we wish to recover the vector field $u_t(x)$, defined by (9), that generates the probability path $p_t(x)$. Exact computation via (9) is generally intractable, as the denominator $p_t(x)$ is defined by an integral (8) that may be difficult to evaluate. Instead, we develop an unbiased stochastic objective for regression of a learned vector field to $u_t(x)$, which generalizes the unconditional flow matching objective (3).

Let $v_\theta(\cdot,\cdot): [0,1] \times \mathbb{R}^d \to \mathbb{R}^d$ be a time-dependent vector field parametrized as a neural network with weights $\theta$. Define the **conditional flow matching (CFM)** objective:

$$\mathcal{L}_{\text{CFM}}(\theta) := \mathbb{E}_{t,q(z),p_t(x|z)}\|v_\theta(t,x) - u_t(x|z)\|^2. \tag{10}$$

The CFM objective describes how to regress against the marginal vector field $u_t(x)$ given by (9) with access only to samples from the conditional probability path $p_t(x|z)$ and conditional vector fields $u_t(x|z)$. This is formalized in the following theorem.

**Theorem 3.2.** *If $p_t(x) > 0$ for all $x \in \mathbb{R}^d$ and $t \in [0,1]$, then, up to a constant independent of $\theta$, $\mathcal{L}_{\text{CFM}}$ and $\mathcal{L}_{\text{FM}}$ are equal, and hence*

$$\nabla_\theta \mathcal{L}_{\text{FM}}(\theta) = \nabla_\theta \mathcal{L}_{\text{CFM}}(\theta). \tag{11}$$

The CFM objective is useful when the marginal vector field $u_t(x)$ is intractable but the conditional vector field $u_t(x|z)$ is tractable. As long as we can efficiently sample from $q(z)$ and $p_t(x|z)$ and calculate $u_t(x|z)$, we can use this stochastic objective to regress $v_\theta$ to the marginal vector field $u_t(x)$.

We discuss the variance arising from the stochastic regression target, and ways to reduce it, in §C.1, Proposition B.2, Proposition B.3.

### 3.2 Sources of conditional probability paths

In this section, we introduce several forms of CFM depending on the choices of $q(z)$, $p_t(\cdot|z)$, and $u_t(\cdot|z)$. All of the CFM variants and related objectives from prior work are summarized in Table 1.

- §3.2.1: We interpret the algorithm of Lipman et al. (2023) (FM from a Gaussian) as a special case of CFM.
- §3.2.2: We relax the Gaussian source requirement by letting the condition $z$ be a pair $(x_0, x_1)$ of an initial and a terminal point. In the basic form of CFM (I-CFM), we take the distribution $q(z)$ to equal $q(x_0)q(x_1)$, allowing generative modeling with an arbitrary source distribution.

---

**Algorithm 1** Conditional Flow Matching

---

**Input:** Efficiently samplable $q(z)$, $p_t(x|z)$, and computable $u_t(x|z)$ and initial network $v_\theta$.
**while** Training **do**
$\quad z \sim q(z); \quad t \sim \mathcal{U}(0,1); \quad x \sim p_t(x|z)$
$\quad \mathcal{L}_{\mathrm{CFM}}(\theta) \leftarrow \|v_\theta(t,x) - u_t(x|z)\|^2$
$\quad \theta \leftarrow \mathrm{Update}(\theta, \nabla_\theta \mathcal{L}_{\mathrm{CFM}}(\theta))$
**return** $v_\theta$

---

Table 1: Probability path definitions for existing methods which fit in the generalized conditional flow matching framework (top) and our newly defined paths (bottom). We define two new probability path objectives that can handle general source distributions and optimal transport flows.

| Probability Path | $q(z)$ | $\mu_t(z)$ | $\sigma_t$ | Cond. OT | Marginal OT | General source |
|---|---|---|---|---|---|---|
| Var. Exploding (Song & Ermon, 2019) | $q(x_1)$ | $x_1$ | $\sigma_{1-t}$ | $\times$ | $\times$ | $\times$ |
| Var. Preserving (Ho et al., 2020) | $q(x_1)$ | $\alpha_{1-t}x_1$ | $\sqrt{1-\alpha_{1-t}^2}$ | $\times$ | $\times$ | $\times$ |
| Flow Matching (Lipman et al., 2023) | $q(x_1)$ | $tx_1$ | $t\sigma - t + 1$ | $\checkmark$ | $\times$ | $\times$ |
| Rectified Flow Liu (2022) | $q(x_0)q(x_1)$ | $tx_1 + (1-t)x_0$ | $0$ | $\checkmark$ | $\times$ | $\checkmark$ |
| Var. Pres. Stochastic Interpolant Albergo & Vanden-Eijnden (2023) | $q(x_0)q(x_1)$ | $\cos(\frac{1}{2}\pi t)x_0 + \sin(\frac{1}{2}\pi t)x_1$ | $0$ | $\checkmark$ | $\times$ | $\checkmark$ |
| Independent CFM | $q(x_0)q(x_1)$ | $tx_1 + (1-t)x_0$ | $\sigma$ | $\checkmark$ | $\times$ | $\checkmark$ |
| (Ours) Optimal Transport CFM | $\pi(x_0, x_1)$ | $tx_1 + (1-t)x_0$ | $\sigma$ | $\checkmark$ | $\checkmark$ | $\checkmark$ |
| (Ours) Schrödinger Bridge CFM | $\pi_{2\sigma^2}(x_0, x_1)$ | $tx_1 + (1-t)x_0$ | $\sigma\sqrt{t(1-t)}$ | $\checkmark$ | $\checkmark$ | $\checkmark$ |

- §3.2.3: We consider joint distributions $q(z) = q(x_0, x_1)$ that are given by minibatch optimal transport maps, causing the learned flow to be an (approximate) OT flow.
- §3.2.4: we consider $q(z)$ given by an entropy-regularized OT map and show that the CFM objective with this $q(z)$ solves the Schrödinger bridge problem.

### 3.2.1 FM from the Gaussian

Lipman et al. (2023) considered the problem of unconditional generative modeling given a training dataset. Identifying the condition $z$ with a single datapoint $z := x_1$, and choosing a smoothing constant $\sigma > 0$, one sets

$$p_t(x|z) = \mathcal{N}(x \mid tx_1, (t\sigma - t + 1)^2), \tag{12}$$

$$u_t(x|z) = \frac{x_1 - (1-\sigma)x}{1 - (1-\sigma)t}, \tag{13}$$

which is a probability path from the standard normal distribution ($p_0(x|z) = \mathcal{N}(x; 0, \boldsymbol{I})$) to a Gaussian distribution centered at $x_1$ with standard deviation $\sigma$ ($p_1(x|z) = \mathcal{N}(x; x_1, \sigma^2)$). If one sets $q(z) = q(x_1)$ to be the uniform distribution over the training dataset, the objective introduced by Lipman et al. (2023) is equivalent to the CFM objective (10) for this conditional probability path.

We emphasize that although the *conditional* probability path $p_t(x|z)$ is an optimal transport path from $p_0(x|z)$ to $p_1(x|z)$, the *marginal* path $p_t(x)$ is **not** in general an OT path from the standard normal $p_0(x)$ to the data distribution $p_1(x)$.

### 3.2.2 Basic form of CFM: Independent coupling

In the basic form of CFM (I-CFM), we identify $z$ with a pair of random variables, a source point $x_0$ and a target point $x_1$, and set $q(z) = q(x_0)q(x_1)$ to be the independent coupling. We let the conditionals be Gaussian flows between $x_0$ and $x_1$ with standard deviation $\sigma$, defined by

$$p_t(x|z) = \mathcal{N}(x \mid tx_1 + (1-t)x_0, \sigma^2), \tag{14}$$

$$u_t(x|z) = x_1 - x_0. \tag{15}$$

We note that the formulation of $u_t(x|z)$ follows from an application of Theorem 2.1 to the conditional probability path with $\mu_t = tx_1 + (1-t)x_0$ and $\sigma_t = \sigma$. Furthermore, we note that $p_t(x|z)$ is efficiently

samplable and $u_t$ is efficiently computable, thus gradient descent on $\mathcal{L}_{\mathrm{CFM}}$ is also efficient. For this choice of $z$, $p_t(\cdot|z)$, and $u_t(\cdot|z)$, we know the marginal boundary probabilities approach $q_0$ and $q_1$ respectively as $\sigma \to 0$. This is made explicit in the following Proposition:

**Proposition 3.3.** *The marginal $p_t$ corresponding to $q(z) = q(x_0)q(x_1)$ and the $p_t(x|z), u_t(x|z)$ in (14) and (15) has boundary conditions $p_1 = q_1 * \mathcal{N}(x \mid 0, \sigma^2)$ and $p_0 = q_0 * \mathcal{N}(x \mid 0, \sigma^2)$, where $*$ denotes the convolution operator.*

In particular, as $\sigma \to 0$, the marginal vector field $u_t(x)$ approaches one that transports the distribution $q(x_0)$ to $q(x_1)$ and can thus be seen as a generative model of $x_1$. Note that there is no requirement for $q(x_0)$ to be Gaussian. Conditioning on $x_0$ allows us to generalize flow matching to arbitrary source distributions with intractable densities. In the case of FM from a Gaussian, while each conditional flow is the dynamic optimal transport flow from $\mathcal{N}(x_0, \sigma^2)$ to $\mathcal{N}(x_1, \sigma^2)$, the marginal vector field $u_t(x)$ is not necessarily an OT flow.

**Connection with related Rectified Flow and Stochastic interpolants methods.** We note that I-CFM is closely related to the algorithms proposed by Albergo & Vanden-Eijnden (2023); Liu (2022). In the case where the conditional probability path $p_t$ is a Dirac (*i.e.,* $\sigma = 0$), I-CFM is equivalent to (Liu, 2022). Furthermore, if we consider the Gaussian mean $\mu_t = \cos(\frac{1}{2}\pi t)x_0 + \sin(\frac{1}{2}\pi t)x_1$ instead of the linear interpolation, I-CFM would be equivalent to the variance preserving stochastic interpolant in Albergo & Vanden-Eijnden (2023), which has also been further generalized.

**Connection to FM from the Gaussian.** There exists a set of conditional probability paths conditioned on $x_1$ *and $x_0 \sim \mathcal{N}(0, 1)$* that have an equivalent probability flow to the marginal $p_t$ of flow matching from the Gaussian (§3.2.1), which is only conditioned on $x_1$. These paths are defined by

$$p_t(x|z) = \mathcal{N}(x \mid tx_1 + (1-t)x_0, (\sigma t)^2 + 2\sigma t(1-t)). \tag{16}$$

Proposition B.1 states an equivalence between I-CFM with these paths and the objective from §3.2.1.

### 3.2.3 Optimal transport CFM

In this section, we present our second main contribution. The formulation in the previous section can readily be generalized to distributions $q(z) = q(x_0, x_1)$ in which $x_0$ and $x_1$ are not independent, as long as $q(z)$ has marginals $q(x_0)$ and $q(x_1)$. Therefore, we propose to set $q(z)$ to be the 2-Wasserstein optimal transport map $\pi$ achieving the infimum in (6), namely,

$$q(z) \coloneqq \pi(x_0, x_1). \tag{17}$$

In this case, $z$ is still a tuple of points, but instead of $x_0, x_1$ being sampled independently from their marginal distributions, they are sampled jointly according to the optimal transport map $\pi$. We call this method *optimal transport CFM* (OT-CFM). If one uses the $p_t(x|z)$ defined by (14) and $u_t(x|z)$ in (15), OT-CFM is equivalent to *dynamic* optimal transport in the following sense.

**Proposition 3.4.** *The results of Proposition 3.3 also hold for $q(z)$ in (17). Furthermore, assuming regularity properties of $q_0$, $q_1$, and the optimal transport plan $\pi$, as $\sigma^2 \to 0$ the marginal path $p_t$ and field $u_t$ minimize (7), i.e., $u_t$ solves the dynamic optimal transport problem between $q_0$ and $q_1$.*

We consider two cases: (1) when the data set is small enough and we know the static optimal transport plan (e.g. single cell data). (2) when the data is too large (or continuous) (e.g. image data) and the static OT plan is computationally infeasible to determine exactly. In the first case we are able to extend the transport map to unseen data similar to the task presented in Bunne et al. (2023). In the second case we show an approximation with minibatch OT improves over a random plan in terms of generative modelling performance and training time.

**Minibatch OT approximation.** For large datasets, the transport plan $\pi$ can be difficult to compute and store due to OT's cubic time and quadratic memory complexity in the number of samples (Cuturi, 2013; Tong et al., 2020). Therefore, we rely on a minibatch OT approximation similar to Fatras et al. (2021b). Although minibatch OT incurs an error relative to the exact OT solution, it has been successfully used in

many applications like domain adaptation or generative modeling (Damodaran et al., 2018; Genevay et al., 2018). Specifically, for each batch of data $(\{x_0^{(i)}\}_{i=1}^B, \{x_1^{(i)}\}_{i=1}^B)$ seen during training, we sample pairs of points from the joint distribution $\pi_{\text{batch}}$ given by the OT plan between the source and target points in the batch. (The OT batch size need not match the optimization batch size, but we keep them equal for simplicity.) Thus, we solve a minibatch approximation of dynamic optimal transport. However, when the OT batch size equals the support size of $(q_0, q_1)$, we recover exact OT and therefore, by Proposition 3.4, learn the exact dynamic optimal transport. We show empirically that the batch size can be much smaller than the full dataset size and still give good performance, which aligns with prior studies (Fatras et al., 2020; 2021a). Concurrently, a similar framework and theoretical results appeared in Pooladian et al. (2023).

### 3.2.4 Schrödinger bridge CFM

Recently, there has been significant effort in learning diffusion models with general source distributions, formulated as a Schrödinger bridge problem (De Bortoli et al., 2021; Vargas et al., 2021; Chen et al., 2022) or bridge matching Peluchetti (2023); Liu et al. (2022b); Ye et al. (2022). Here we show that SB-CFM, an entropic variant of OT-CFM, can be used to train an ODE to match the probability flow of a Schrödinger bridge with a Brownian motion reference process.

Let $p_{\text{ref}}$ be the standard Wiener process scaled by $\sigma$ with initial-time marginal $p_{\text{ref}}(x_0) = q(x_0)$. The Schrödinger bridge problem (Schrödinger, 1932) seeks the process $\pi$ that is closest to $p_{\text{ref}}$ while having initial and terminal marginal distributions specified by the data distribution $q(x_0)$ and $q(x_1)$:

$$\pi^* := \underset{\pi(x_0)=q(x_0),\pi(x_1)=q(x_1)}{\arg\min} \text{KL}(\pi \,\|\, p_{\text{ref}}). \tag{18}$$

We define the joint distribution

$$q(z) \coloneqq \pi_{2\sigma^2}(x_0, x_1) \tag{19}$$

where $\pi_{2\sigma^2}$ is the solution of the entropy-regularized optimal transport problem (Cuturi, 2013) with cost $\|x_0 - x_1\|$ and entropy regularization $\lambda = 2\sigma^2$ (see (33) for the background on entropic OT). We set the conditional path distribution to be a Brownian bridge with diffusion scale $\sigma$ between $x_0$ and $x_1$, with probability path and generating vector field

$$p_t(x \mid z) = \mathcal{N}(x \mid tx_1 + (1-t)x_0, t(1-t)\sigma^2) \tag{20}$$

$$u_t(x \mid z) = \frac{1-2t}{2t(1-t)}(x - (tx_1 + (1-t)x_0)) + (x_1 - x_0), \tag{21}$$

where $u_t$ is computed by (5) as the vector field generating the probability path $p_t(x|z)$. The marginal coupling $\pi_{2\sigma^2}$ and $u_t(x|z)$ define $u_t(x)$, which is approximated by the regression objective in Alg. 4. The solution of the SB is known to be the map which is the solution of the entropically-regularized OT problem, motivating the next proposition.

**Proposition 3.5.** *The marginal vector field $u_t(x)$ defined by (19) and (21) generates the same marginal probability path as the solution $\pi^*$ to the SB problem in (18).*

While we define SB-CFM with an entropic regularization coefficient of $\varepsilon = 2\sigma^2$, the flow still matches the marginals for any choice of $\varepsilon$. Interestingly, we recover OT-CFM when $\varepsilon \to 0$ and I-CFM when $\varepsilon \to \infty$. A similar result was proven in a concurrent work Pooladian et al. (2023).

## 4 Related work

**Simulation-free continuous-time modeling.** Simulation-free training is common in stochastic flow models where backpropagating through the simulation is numerically challenging and has high variance (Li et al., 2020). While these diffusion models have recently achieved exceptional generative performance on many tasks (Sohl-Dickstein et al., 2015; Song & Ermon, 2019; 2020; Ho et al., 2020; Song et al., 2021b; Dhariwal & Nichol, 2021; Watson et al., 2022b), their simulation requires an inherently costly SDE simulation with many follow-up works to improve inference efficiency (Lu et al., 2022; Salimans & Ho, 2022; Watson et al., 2022a;

Table 2: Comparison of neural optimal transport methods over four distribution pairs ($\mu \pm \sigma$ over five seeds) in terms of fit (2-Wasserstein), optimal transport performance (normalized path energy), and runtime. '—' indicates a method that requires a Gaussian source. Best in **bold**. CFM and RF models are trained on a single CPU core, other baselines are trained with a GPU and two CPUs.

| Dataset → | $\mathcal{N}$→8gaussians | | moons→8gaussians | | $\mathcal{N}$→moons | | $\mathcal{N}$→scurve | | Avg. train time |
|---|---|---|---|---|---|---|---|---|---|
| Algorithm ↓ Metric → | $W_2^2$ | NPE | $W_2^2$ | NPE | $W_2^2$ | NPE | $W_2^2$ | NPE | ($\times 10^3$ s) |
| OT-CFM | $1.262_{\pm0.348}$ | $\mathbf{0.018}_{\pm0.014}$ | $\mathbf{1.923}_{\pm0.391}$ | $\mathbf{0.053}_{\pm0.035}$ | $\mathbf{0.239}_{\pm0.048}$ | $0.087_{\pm0.061}$ | $\mathbf{0.264}_{\pm0.093}$ | $\mathbf{0.027}_{\pm0.026}$ | $1.129_{\pm0.335}$ |
| I-CFM | $1.284_{\pm0.384}$ | $0.222_{\pm0.032}$ | $1.977_{\pm0.266}$ | $2.738_{\pm0.181}$ | $0.338_{\pm0.109}$ | $0.841_{\pm0.148}$ | $0.333_{\pm0.060}$ | $0.867_{\pm0.117}$ | $\mathbf{0.630}_{\pm0.365}$ |
| 2-RF (Liu, 2022) | $1.436_{\pm0.344}$ | $0.069_{\pm0.027}$ | $2.211_{\pm0.423}$ | $0.149_{\pm0.101}$ | $0.278_{\pm0.026}$ | $\mathbf{0.076}_{\pm0.067}$ | $0.395_{\pm0.111}$ | $0.112_{\pm0.085}$ | $0.862_{\pm0.166}$ |
| 3-RF (Liu, 2022) | $1.337_{\pm0.367}$ | $0.055_{\pm0.043}$ | $2.700_{\pm0.587}$ | $0.123_{\pm0.112}$ | $0.305_{\pm0.026}$ | $0.084_{\pm0.051}$ | $0.395_{\pm0.082}$ | $0.129_{\pm0.075}$ | $0.954_{\pm0.116}$ |
| FM (Lipman et al., 2023) | $1.062_{\pm0.196}$ | $0.174_{\pm0.030}$ | — | — | $0.246_{\pm0.077}$ | $0.778_{\pm0.144}$ | $0.377_{\pm0.099}$ | $0.772_{\pm0.081}$ | $0.708_{\pm0.370}$ |
| Reg. CNF (Finlay et al., 2020) | $1.144_{\pm0.075}$ | $0.274_{\pm0.060}$ | — | — | $0.376_{\pm0.040}$ | $0.620_{\pm0.088}$ | $0.581_{\pm0.195}$ | $0.586_{\pm0.503}$ | $8.021_{\pm3.288}$ |
| CNF (Chen et al., 2018) | $\mathbf{1.055}_{\pm0.059}$ | $0.151_{\pm0.064}$ | — | — | $0.387_{\pm0.065}$ | $2.937_{\pm1.973}$ | $0.645_{\pm0.343}$ | $10.548_{\pm8.100}$ | $18.810_{\pm12.677}$ |
| ICNN (Makkuva et al., 2020) | $1.771_{\pm0.398}$ | $0.747_{\pm0.029}$ | $2.193_{\pm0.136}$ | $0.832_{\pm0.004}$ | $0.532_{\pm0.046}$ | $0.267_{\pm0.010}$ | $0.753_{\pm0.068}$ | $0.344_{\pm0.045}$ | $2.912_{\pm0.626}$ |

Song et al., 2021a; Bao et al., 2022). These methods generally consider a simple Gaussian diffusion process, and do not consider generalizing the source distribution. Other works consider general source distributions but this makes optimization and inference more challenging, needing multiple iterations or other tricks to perform well (Wang et al., 2021; De Bortoli et al., 2021; Vargas et al., 2021).

Prior work considering simulation-free training of CNFs considers algorithms that are equivalent to CFM with Gaussian source distribution (Rozen et al., 2021; Ben-Hamu et al., 2022; Lipman et al., 2023) or independent samples from $q_0, q_1$ (Albergo & Vanden-Eijnden, 2023; Albergo et al., 2023; Neklyudov et al., 2023). Recent work also studies Schrödinger bridges from unpaired samples (Shi et al., 2022) and regularization of flows using dynamic OT (Liu et al., 2023b). We also note the work Pooladian et al. (2023), concurrent with the preprint version of this paper. Other concurrent works explore various solutions to approximate Schrödinger bridges (Somnath et al., 2023; Shi et al., 2023; Liu et al., 2023a).

**Dynamic optimal transport.** There are a variety of methods that consider dynamic OT between continuous distributions with neural networks; however, these require constrained architectures (Leygonie et al., 2019; Makkuva et al., 2020; Bunne et al., 2022) or use a regularized CNF, which is challenging to optimize (Tong et al., 2020; Finlay et al., 2020; Onken et al., 2021; Huguet et al., 2022a). With our work it is possible to achieve optimal transport flows without either of these constraints.

## 5 Experiments

In this section we empirically evaluate the I-CFM, OT-CFM, and SB-CFM objectives, as well as algorithms from prior work, with respect to both optimal transport and generative modeling criteria. All experiment details can be found in Appendix E.

### 5.1 Low-dimensional data: Optimal transport and faster convergence

We evaluate how well various models perform dynamic optimal transport and generative modeling in low dimensions. We train ODEs mapping between four pairs of two-dimensional datasets: between a standard Gaussian and 8gaussians, moons, and scurve and between moons and 8gaussians.

**OT-CFM approximates dynamic OT.** To measure how well a model solves the OT problem we use normalized path energy (NPE), defined via the 2-Wasserstein distance as $\mathrm{NPE}(v_\theta) = |\mathrm{PE}(v_\theta) - W_2^2(q_0, q_1)|/W_2^2(q_0, q_1)$, where the path energy (PE) is $\mathrm{PE}(v_\theta) = \mathbb{E}_{x(0)\sim q(x_0)} \int_0^1 \|v_\theta(t, x(t))\|^2 dt$. Table 2 summarizes our results showing that OT-CFM flows generalize better to the test set and are very close to the dynamic OT paths as measured by normalized path energy. We find transforming moons↔8gaussians to be particularly challenging to learn for I-CFM as compared to OT-CFM; the learned paths are depicted in Fig. 1 (bottom). Although OT-CFM uses a minibatch OT map, we find that OT-CFM requires surprisingly small batches to approximate the OT map well, suggesting some generalization advantages of the network optimization (Fig. D.2).

Figure 2: **Left:** OT-CFM trains faster, in terms of validation set error, than CFM and FM models. **Right:** With different ODE integrators, OT-CFM reduces the error for a fixed number of function evaluations during inference.

Table 3: Schrödinger bridge flow comparison, showing average error over flow time to ground truth averaged over 5 models for SB-CFM and 5 dynamics from DSB (De Bortoli et al., 2021).

| Dataset ↓ Alg. → | SB-CFM | DSB |
|---|---|---|
| $\mathcal{N}$→8gaussians | **0.454 $\pm$ 0.164** | 1.440 $\pm$ 0.720 |
| moons→8gaussians | **1.377 $\pm$ 0.229** | 2.407 $\pm$ 1.025 |
| $\mathcal{N}$→moons | **0.283 $\pm$ 0.048** | 0.333 $\pm$ 0.129 |
| $\mathcal{N}$→scurve | **0.297 $\pm$ 0.064** | 0.383 $\pm$ 0.134 |

**OT-CFM yields faster training.** By conditioning on minibatch optimal transport flows, OT-CFM is substantially easier to train, which we posit is due to the variance reduction of the conditional flow. In Fig. 2 (left), we evaluate the performance over time of OT-CFM against CFM and FM objectives. For the same number of steps OT-CFM has better performance on the validation set. In Table D.1, we compare the training times for various Neural OT methods whose performance can be seen in Table 2. Simulation-free optimization is significantly faster to train with equal or superior performance.

**OT-CFM yields faster inference.** We next evaluate the quality of samples during inference time. In Fig. 2 (right), we compare the quality of samples for different number of function evaluations (NFEs) across different flow matching objectives. In this experiment we sample from the source distribution test set and simulate the ODE over time for different solvers. We find that OT-CFM consistently requires fewer evaluations to achieve the same quality and achieves better quality with the same NFEs. This is consistent with previous work, which found OT paths lead to faster, higher quality inference in regularized CNFs (Finlay et al., 2020; Onken et al., 2021) and flow matching vs. standard variance-preserving and variance-exploding probability paths (Lipman et al., 2023).

**SB-CFM reproduces Schrödinger bridge flows.** There are a number of methods which theoretically converge to a Schrödinger bridge between two datasets. In Table 3 we compare SB-CFM and the diffusion Schrödinger bridge (DSB) method introduced in De Bortoli et al. (2021) on the quality of the learnt Schrödinger bridges based on the average 2-Wasserstein distance to ground truth Schrödinger bridge samples over 18 time steps. Furthermore, SB-CFM is also significantly faster than DSB (Table D.1).

## 5.2 Application to single-cell interpolation

As a specific application, we consider the task of single-cell trajectory interpolation. In this task we use leave-one-out validation over the timepoints. From times data at times $[0, t-1], [t+1, T]$ we try to interpolate its distribution at time $t$ following the setup of Schiebinger et al. (2019); Tong et al. (2020); Huguet et al. (2022a). Low error means we model individual cells well, which is useful in a number of downstream tasks such as gene regulatory network inference (Aliee et al., 2021; Yeo et al., 2021). Following Huguet et al. (2022b), we repurpose the CITE-seq and Multiome datasets from a recent NeurIPS competition for this task (Burkhardt et al., 2022). We also include the Embroid body data from Moon et al. (2019); Tong et al. (2020). Table 4 shows the average earth mover's distance (1-Wasserstein) on left–out timepoints for three datasets. On all three datasets OT-CFM outperforms other methods and baselines on average.

Table 4: Single-cell comparison over three datasets averaged over leaving out intermediate timepoints measuring EMD to left out distribution following Tong et al. (2020). *Indicates values taken from aforementioned work.

| Algorithm ↓ Dataset → | Cite | EB | Multi |
|---|---|---|---|
| TrajectoryNet (Tong et al., 2020)* | — | $0.848 \pm$ — | — |
| Reg. CNF (Finlay et al., 2020)* | — | $0.825 \pm$ — | — |
| DSB (De Bortoli et al., 2021) | $0.953 \pm 0.140$ | $0.862 \pm 0.023$ | $1.079 \pm 0.117$ |
| I-CFM | $0.965 \pm 0.111$ | $0.872 \pm 0.087$ | $1.085 \pm 0.099$ |
| SB-CFM | $1.067 \pm 0.107$ | $1.221 \pm 0.380$ | $1.129 \pm 0.363$ |
| OT-CFM | $\mathbf{0.882 \pm 0.058}$ | $\mathbf{0.790 \pm 0.068}$ | $\mathbf{0.937 \pm 0.054}$ |

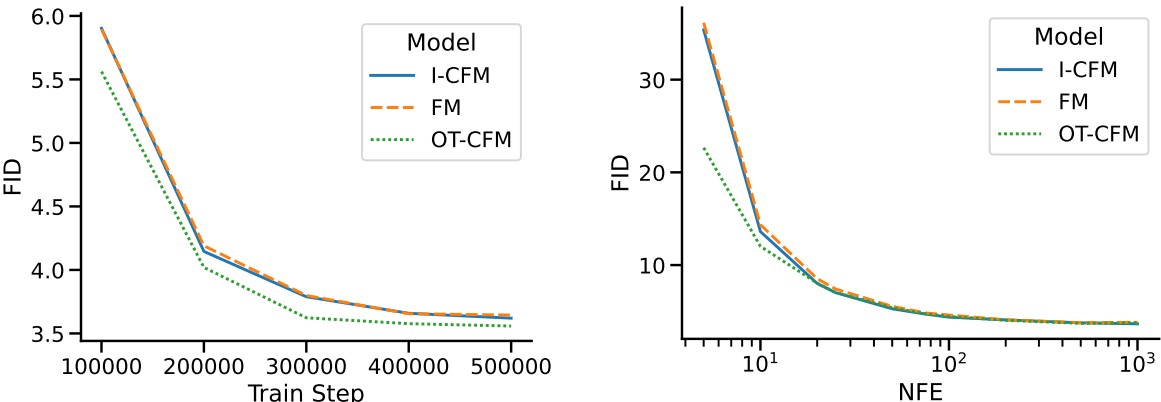

Figure 3: **Left:** Fréchet inception distance (FID) scores on CIFAR-10 for different numbers of training steps using a DOPRI5 adaptive solver. **Right:** FID scores on CIFAR-10 using Euler integration for various numbers of function evaluations (NFE) per sample after 400k training steps. In both cases, OT-CFM outperforms I-CFM and FM models, showing the benefits of minibatch optimal transport.

## 5.3 High-dimensional data: Lower-cost training and inference

We perform an experiment on unconditional CIFAR-10 generation from a Gaussian source to examine how OT-CFM performs in the high-dimensional image setting. We use a similar setup to that of Lipman et al. (2023), including the time-dependent U-Net architecture from Nichol & Dhariwal (2021) that is commonly used in diffusion models. We were not able to reproduce the results reported from Lipman et al. (2023) with the parameters specified in the paper.[3] Therefore, we selected different training hyperparameters.

The main differences with Lipman et al. (2023) are that we use a constant learning rate, set to $2 \times 10^{-4}$, instead of a linearly decreasing one (from $5 \times 10^{-4}$ to $10^{-8}$). To prevent training instabilities and variance, we clip the gradient norm to 1 and rely on exponential moving average with a decay of 0.9999. Regarding the architecture, we used the same as Lipman et al. (2023), but with a smaller number of channels (128 instead of 256), leading to much faster training, as well as 10% dropout. Furthermore, our batch size was 128 instead of 256, which leads to a reduced memory cost.

We train our OT-CFM, as well as I-CFM and the original FM, with this new training procedure and report the Fréchet inception distance (FID) in Table 5. In Fig. 3 (left), we show the FID over training time with the Dormand-Prince fifth-order adaptive solver (Hairer et al., 1993, DOPRI5) using a relative and absolute error threshold of $10^{-5}$ similarly to Lipman et al. (2023), and in Fig. 3 (right), we present the FID as a function of the numbers of function evaluations (NFE) using Euler integration.

We find that:

---

[3]Specifically, we find that the number generated samples for FID calculation, the value of the smoothing constant $\sigma_{\min}$, any data augmentation used, the standard deviation of the distribution $p_0$, and the batch size used during evaluation (which can affect the function evaluation count with adaptive integrators) are not specified in Lipman et al. (2023)'s manuscript. In addition, contradictory information is given about the number of training epochs.

Table 5: FID score and number of function evaluations (NFE) for different ODE solvers: fixed-step Euler integration with 100 and 1000 steps and adaptive integration (DOPRI5). The adaptive solver is significantly better than the Euler solver in fewer steps. First three results are from Lipman et al. (2023) and fourth from Albergo & Vanden-Eijnden (2023). The fifth line is our reproduced results following Lipman et al. (2023)'s training procedure. We have run OT-FM, S.I. and VP-FM following our training procedure and we have denoted them (ours). The two last rows report the results of our proposed methos I-CFM and OT-CFM.

| NFE / sample → | 100 | 1000 | Adaptive | |
| --- | --- | --- | --- | --- |
| Algorithm ↓ | FID | FID | FID | NFE |
| DDPM | | | 7.48 | 274 |
| OT-FM (reported) | | | 6.35 | 142 |
| VP-FM (reported) | | | 8.06 | 183 |
| S.I. (reported) | | | 10.27 | |
| OT-FM (reproduced) | 13.742 | 12.491 | 11.527 | 139.83 |
| VP-FM (ours) | 7.772 | 4.048 | 4.335 | 525.92 |
| OT-FM (ours) | 4.640 | 3.822 | 3.655 | 143.00 |
| S.I. (ours) | 4.488 | 4.132 | 4.009 | 146.12 |
| I-CFM (ours) | 4.461 | **3.643** | 3.659 | 146.42 |
| OT-CFM (ours) | **4.443** | 3.741 | **3.577** | **133.94** |

- With improved hyperparameters, we achieve a significantly better FID with the FM training objective than the one reported by Lipman et al. (2023) at a lower cost.
- For a short computation budget, OT-CFM outperforms FM and (non-OT) I-CFM (Table 5, left).
- After a long training time, all methods achieve similar performance at a high number of function evaluations using fixed-step ODE integration, but OT-CFM performs significantly better with a small number of function evaluations (*i.e.*, allows more efficient inference), indicating straighter, easily integrable flows (Table 5, right).
- FM and I-CFM are equivalently computationally efficient per iteration and OT-CFM comes with a low (<1%) computational overhead during training.

Our new training procedures, available at `https://github.com/atong01/conditional-flow-matching`, allow us to outperform the previous reported results from Lipman et al. (2023), while the results with our OT-CFM are state-of-the-art for simulation-free neural ODE training algorithms.

### 5.4 OT-CFM for unsupervised translation

We show how CFM can be used to learn a mapping between two unpaired datasets in high-dimensional space using the CelebA dataset (Liu et al., 2015; Sun et al., 2014), which consists of $\sim 200$k images of faces together with 40 binary attribute annotations. For each attribute, we wish to learn an invertible mapping between images with and without the attribute (*e.g.*, 'not smiling'↔'smiling').

To reduce dimensionality, we first train a VAE on the images and encode them as 128-dimensional latent vectors. For each attribute, we learn a flow to map between the embeddings of images without the attribute and those of images with the attribute. After the CNF is learned, we push forward a held-out set of negative vectors by the CNF and compare them to the held-out positive vectors and vice versa. As a metric of divergence, we use maximum mean discrepancy (MMD) with a broad Gaussian kernel ($\exp(-\|x - y\|^2/(2 \cdot 128))$). The results aggregated over all attributes are shown in Table 6, showing that OT-CFM discovers a better mapping than other methods. Although MMD is lower for larger $\sigma$, we found that the alignment is less natural when $\sigma$ is large, and performance begins to degrade when $\sigma > 1$. Fig. E.1 shows several visualizations of the learned trajectories.

Finally, while here we work in a latent space, future work should consider learning flows directly in image space, where GAN-based approaches (Zhu et al., 2017) continue to dominate.

### 5.5 Additional experiments and extensions

We present numerous other extensions, applications, and evaluations of CFM in Appendix D, notably:

Table 6: MMD (in units of $10^{-3}$) between target and transformed source samples of CelebA latent vectors. Mean and standard deviation over 40 attributes and both translation directions $(- \leftrightarrow +)$ for each attribute. 'Identity' refers to performing no translation and treating source samples as approximate samples from the target.

| Algorithm ↓ | $\sigma = 0.1$ | $\sigma = 0.3$ | $\sigma = 1$ |
|---|---|---|---|
| Identity | $9.17 \pm 5.68$ | $9.17 \pm 5.68$ | $9.17 \pm 5.68$ |
| I-CFM | $4.85 \pm 5.09$ | $3.44 \pm 2.03$ | $1.59 \pm 0.83$ |
| OT-CFM | $2.81 \pm 2.62$ | $1.91 \pm 1.30$ | $1.04 \pm 0.60$ |

**OT-CFM reduces variance in the regression target.** To accompany the theoretical results in §C.1, in §D.1 we empirically study the variance of the stochastic regression objective in (OT-)CFM. The results suggest an explanation for the faster convergence of models trained with OT-CFM.

**Energy-based CFM.** In §C.2 and §D.2 we show how CFM can be used to fit samplers for unnormalized density functions, where exact samples from $q(x_0)$ or $q(x_1)$ are not available.

**Extension to stochastic dynamics.** Tong et al. (2024) extends CFM to allow learning *stochastic* dynamics from unpaired source and target data.

## 6 Conclusion

We have introduced a novel class of simulation-free objectives for learning continuous-time flows with a general source distribution. Our approach to training continuous normalizing flows and conditional flow models does not require integration over time during training. We have shown that lifting the static optimal transport problem to the dynamic setting leads to simulation-free solutions to the dynamic OT and SB problems, while also allowing more efficient training and inference of flow models by lowering variance of the objective and straightening flows. One limitation of CFM is that it requires closed-form conditional flows, which hinders its application to situations where we want to regularize the marginal vector field $u_t(x)$ based on prior information (Tong et al., 2020). In addition, the minibatch approximation to OT can incur error in high dimensions; subsequent work can consider the use of neural-network approximations to OT maps (Korotin et al., 2023b;a) in conjunction with CFM. We expect future work to overcome these limitations and hope that ideas from conditional flow matching will improve high-dimensional generative models.

## Contribution statement

A.T. initially conceived the idea. Y.Z., G.H., and N.M. led the development of the theory. High-dimensional experiments and open-source code were led by A.T. and K.F. Additional experiments were contributed by Y.Z., J.R., G.H., N.M., and A.T. All authors contributed to designing the experiments. N.M. and A.T. drove the writing of the paper, with contributions from all other authors. G.W. and Y.B. guided the project.

## Acknowledgments

We would like to thank Stefano Massaroli for productive conversations as well as thank Xinyu Yuan, Marco Jiralerspong, Tara Akhound-Sadegh and Joey Bose for their helpful comments and feedback on the manuscript. We are also grateful to the anonymous reviewers for suggesting numerous improvements. This research was enabled in part by compute resources provided by Mila (mila.quebec) and NVIDIA Corporation. The authors acknowledge funding from CIFAR, Genentech, Samsung, and IBM. In addition, K.F. acknowledges funding from NSERC (RGPIN-2019-06512) and G.W. acknowledges funding from NSERC Discovery grant 03267 and NIH grant R01GM135929.

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

## A  Proofs of theorems

**Theorem 3.1.** *The marginal vector field (9) generates the probability path (8) from initial conditions $p_0(x)$.*

*Proof of Theorem 3.1.* To verify this, we first check that $p_t$ and $u_t$ satisfy the continuity equation.

We start with the derivative w.r.t. time of (8)

$$\frac{d}{dt}p_t(x) = \frac{d}{dt}\int p_t(x|z)q(z)dz$$

by Leibniz Rule,

$$= \int \frac{d}{dt} \left( p_t(x|z)q(z) \right) dz$$

since $u_t(\cdot|z)$ generates $p_t(\cdot|z)$,

$$= - \int \mathrm{div} \left( u_t(x|z)p_t(x|z)q(z) \right) dz$$

exchanging the derivative and integral,

$$= -\mathrm{div} \left( \int u_t(x|z)p_t(x|z)q(z) dz \right)$$

Using (9),

$$= -\mathrm{div} \left( u_t(x)p_t(x) \right)$$

satisfying the continuity equation $\frac{d}{dt}p_t(x) + \mathrm{div}\left( u_t(x)p_t(x) \right) = 0$. $\qquad \square$

**Theorem 3.2.** *If $p_t(x) > 0$ for all $x \in \mathbb{R}^d$ and $t \in [0,1]$, then, up to a constant independent of $\theta$, $\mathcal{L}_{\mathrm{CFM}}$ and $\mathcal{L}_{\mathrm{FM}}$ are equal, and hence*

$$\nabla_\theta \mathcal{L}_{\mathrm{FM}}(\theta) = \nabla_\theta \mathcal{L}_{\mathrm{CFM}}(\theta). \tag{11}$$

*Proof of Theorem 3.2.* For this proof we need (8), (9) and the existence and exchange of many integrals. As in Lipman et al. (2023) we assume that $q$, $p_t(x|z)$ are decreasing to zero at sufficient speed as $\|x\| \to \infty$ and that $u_t, v_t, \nabla_\theta v_t$ are bounded.

$$\nabla_\theta \mathbb{E}_{p_t(x)} \|v_\theta(t,x) - u_t(x)\|^2 = \nabla_\theta \mathbb{E}_{p_t(x)} \left( \|v_\theta(t,x)\|^2 - 2 \langle v_\theta(t,x), u_t(x) \rangle + \|u_t(x)\|^2 \right)$$
$$= \nabla_\theta \mathbb{E}_{p_t(x)} \left( \|v_\theta(t,x)\|^2 - 2 \langle v_\theta(t,x), u_t(x) \rangle \right)$$

$$\nabla_\theta \mathbb{E}_{q(z),p_t(x|z)} \|v_\theta(t,x) - u_t(x|z)\|^2 =$$
$$\nabla_\theta \mathbb{E}_{q(z),p_t(x|z)} \left( \|v_\theta(t,x)\|^2 - 2 \langle v_\theta(t,x), u_t(x|z) \rangle + \|u_t(x|z)\|^2 \right)$$
$$= \mathbb{E}_{q(z),p_t(x|z)} \nabla_\theta \left( \|v_\theta(t,x)\|^2 - 2 \langle v_\theta(t,x), u_t(x|z) \rangle \right)$$

By bilinearity of the 2-norm and since $u_t$ is independent of $\theta$. Next,

$$\mathbb{E}_{p_t(x)} \|v_\theta(t,x)\|^2 = \int \|v_\theta(t,x)\|^2 p_t(x) dx$$
$$= \iint \|v_\theta(t,x)\|^2 p_t(x|z)q(z) dz dx$$
$$= \mathbb{E}_{q(z),p_t(x|z)} \|v_\theta(t,x)\|^2$$

Finally,

$$\mathbb{E}_{p_t(x)} \langle v_\theta(t,x), u_t(x) \rangle = \int \left\langle v_\theta(t,x), \frac{\int u_t(x|z)p_t(x|z)q(z) dz}{p_t(x)} \right\rangle p_t(x) dx$$
$$= \int \left\langle v_\theta(t,x), \int u_t(x|z)p_t(x|z)q(z) dz \right\rangle dx$$
$$= \iint \langle v_\theta(t,x), u_t(x|z) \rangle p_t(x|z)q(z) dz dx$$
$$= \mathbb{E}_{q(z),p_t(x|z)} \langle v_\theta(t,x), u_t(x|z) \rangle$$

Where we first substitute (9) then change the order of integration for the final equality. Since at all times $t$ the gradients of $\mathcal{L}_{\mathrm{FM}}$ and $\mathcal{L}_{\mathrm{CFM}}$ are equal, $\nabla_\theta \mathcal{L}_{\mathrm{FM}}(\theta) = \nabla_\theta \mathcal{L}_{\mathrm{CFM}}(\theta)$ $\qquad \square$

**Proposition 3.3.** *The marginal $p_t$ corresponding to $q(z) = q(x_0)q(x_1)$ and the $p_t(x|z), u_t(x|z)$ in (14) and (15) has boundary conditions $p_1 = q_1 * \mathcal{N}(x \mid 0, \sigma^2)$ and $p_0 = q_0 * \mathcal{N}(x \mid 0, \sigma^2)$, where $*$ denotes the convolution operator.*

*Proof of Proposition 3.3.* We start with (8) to show the result of the lemma. We note that $q(z) = q((x_0, x_1)) = q(x_0)q(x_1)$

$$
\begin{aligned}
p_t(x) &= \int p_t(x|z)q(z)dz \\
&= \int \mathcal{N}(x\,|tx_1 + (1-t)x_0, \sigma^2)q((x_0, x_1))d(x_0, x_1) \\
&= \iint \mathcal{N}(x\,|tx_1 + (1-t)x_0, \sigma^2)q(x_0)q(x_1)dx_0 dx_1
\end{aligned}
$$

evaluated at $i = 0, 1$ respectively. Therefore, at $t = 0$,

$$
\begin{aligned}
p_0(x) &= \iint \mathcal{N}(x\,|x_0, \sigma^2)q(x_0)q(x_1)dx_0 dx_1 \\
&= \int \mathcal{N}(x\,|x_0, \sigma^2)q(x_0)dx_0 \\
&= q(x_0) * \mathcal{N}(x\,|0, \sigma^2).
\end{aligned}
$$

This is also true for $t = 1$. $\qquad\square$

**Proposition 3.4.** *The results of Proposition 3.3 also hold for $q(z)$ in (17). Furthermore, assuming regularity properties of $q_0$, $q_1$, and the optimal transport plan $\pi$, as $\sigma^2 \to 0$ the marginal path $p_t$ and field $u_t$ minimize (7), i.e., $u_t$ solves the dynamic optimal transport problem between $q_0$ and $q_1$.*

*Proof of Proposition 3.4.* We will assume certain regularity conditions on $q_0$, $q_1$, and $\pi$ to allow reduction to known results. We leave it to future work to determine which of these conditions are necessary and which are redundant with other conditions. However, because we are concerned with approximation of $u_t(x)$ with neural networks, which are typically smooth, results that relax the regularity assumptions may be vacuous in practice.

*Preliminaries.* We assume that $q_0$ and $q_1$ are compactly supported and admit bounded densities with respect to the Lebesgue measure. Then the conditions for Brenier's theorem (Brenier, 1991) are satisfied. By Brenier's theorem, the optimal joint $\pi$ is unique and is supported on the graph $(x, T(x))$ of a Monge map $T : \mathbb{R}^d \to \mathbb{R}^d$, and $p_t(x)$ is equal to McCann's interpolation (Peyré & Cuturi, 2019, Chapter 7)

$$
p_t = ((1-t)\mathrm{Id} + tT)_{\#}p_0. \tag{22}
$$

In addition, we know that $T(x)$ can be parameterized as the gradient of a convex function, *i.e.*, $T(x) = \nabla\psi(x)$. This characterization of $T$ implies that the conditional probability paths, given by $\phi_t(x) = x + t(T(x) - x)$, do not cross, *i.e.*, $p_t(x|x_0, T(x_0)) = p_t(x)$ for all $(t, x)$.[4] It is known that the probability path $p_t = [\phi_t]_{\#}p_0$ and its associated vector field, given by $u_t(\phi_t(x)) = T(x) - x$, solve the optimal transport problem (Benamou & Brenier, 2000, Proposition 1.1).

We assume that the induced marginals $p_t$ have bounded densities with respect to Lebesgue measure and that $T$ is almost everywhere continuous in $x$, which implies the same for $\phi_t$. Injectivity of $\phi_t$ and noncrossing of paths implies $u_t(\phi_t(x))$ is almost everywhere continuous in $\phi_t(x)$.

*Formal statement of the result.* Denote by $p_t^\sigma(x)$, $p_t^\sigma(x|z)$ the densities in Proposition 3.3, and by $p_t^\sigma(x)$. We will show that for $p_t$-almost every $x$ and almost every $t$,

$$
u_t(x) = \lim_{\sigma \to 0} u_t(x) = \lim_{\sigma \to 0} \frac{\mathbb{E}_{z \sim q(z)} p_t^\sigma(x|z) u_t(x|z)}{p_t^\sigma(x)}. \tag{23}
$$

---

[4]Proof: If the paths from distinct $x_0$ and $x_0'$ cross, so $\phi_t(x_0) = \phi_t(x_0')$, then $(1-t)x_0 + t\nabla\psi(x_0) = (1-t)x_0' + t\nabla\psi(x_0')$. Taking dot product with $x_0 - x_0'$, $(t-1)\|x_0 - x_0'\|^2 = t\langle\nabla\psi(x_0) - \nabla\psi(x_0'), x_0 - x_0'\rangle$. However, we have $(t-1)\|x_0 - x_0'\|^2 < 0$ and $t\langle\nabla\psi(x_0) - \nabla\psi(x_0'), x_0 - x_0'\rangle \geq 0$ by convexity of $\psi$, contradiction.

*Proof.* It suffices to show the equality for $x$ in the support of $p_t$, or in the image of the support of $q_0$ under $\phi_t$. We identify $z$ in the expectation with a pair $(x_0, T(x_0))$ due to $q(x_0, x_1) = \pi(x_0, x_1)$ having support on the graph of the Monge map. Noting that

$$u_t\Big(x\big|(x_0, T(x_0))\Big) = u_0(x_0) = T(x_0) - x_0 \quad \forall x,$$

we see that (23) is equivalent to

$$u_0(\phi_t^{-1}(x)) = \lim_{\sigma \to 0} \frac{\mathbb{E}_{q(x_0)} p_t^\sigma\Big(x\big|(x_0, T(x_0))\Big) u_0(x_0)}{p_t^\sigma(x)}. \tag{24}$$

By the same argument as in the proof of Proposition 3.3, we have that $p_t^\sigma = p_t * \mathcal{N}(0, \sigma^2)$, by integration over $x_0$ of the conditional equality $p_t\Big(\cdot\big|(x_0, T(x_0))\Big) = \mathcal{N}(\phi_t(x_0), \sigma^2) = \delta_{\phi_t(x_0)} * \mathcal{N}(0, \sigma^2)$.

Symmetry of the Gaussian implies that

$$p_t^\sigma(x|(x_0, T(x_0))) = p_t^\sigma(\phi_t(x_0)|(\phi_t^{-1}(x), T(\phi_t^{-1}(x)))).$$

Therefore

$$
\begin{aligned}
\mathbb{E}_{q(x_0)} p_t^\sigma(x|(x_0, T(x_0))) u_0(x_0) &= \mathbb{E}_{q(x_0)} \left[ p_t^\sigma(\phi_t(x_0)|(\phi_t^{-1}(x), T(\phi_t^{-1}(x)))) u_0(x_0) \right] \\
\text{[by change of variables } x' = \phi_t(x_0)] &= \mathbb{E}_{p_t(x')} \left[ p_t^\sigma(x'|(\phi_t^{-1}(x), T(\phi_t^{-1}(x)))) u_0(\phi_t^{-1}(x')) \right] \\
&= \mathbb{E}_{p_t^\sigma(x'|(\phi_t^{-1}(x), T(\phi_t^{-1}(x))))} \left[ p_t(x') u_0(\phi_t^{-1}(x')) \right] \\
&= \mathbb{E}_{\Delta x \sim \mathcal{N}(0, \sigma^2)} \left[ p_t(x + \Delta x) u_0(\phi_t^{-1}(x + \Delta x)) \right] \\
&= \left( p_t(\cdot) u_0(\phi_t^{-1}(\cdot)) * \mathcal{N}(0, \sigma^2) \right)(x).
\end{aligned}
$$

The standard fact that $\mathcal{N}(0, \sigma^2) \xrightarrow{\sigma \to 0} \delta_0$ in distribution implies that if $f$ is a bounded, almost everywhere continuous, compactly supported function, then $(f * \mathcal{N})(x) \to f(x)$ pointwise for almost every $x$. By the hypotheses, $p_t$ and $p_t \cdot (u_0 \circ \phi_t^{-1})$ have this property. It follows that, for every $t$ and almost all $x$,

$$
\begin{aligned}
\lim_{\sigma \to 0} \frac{\mathbb{E}_{q(x_0)} p_t^\sigma(x|(x_0, T(x_0))) u_0(x_0)}{p_t^\sigma(x)} &= \lim_{\sigma \to 0} \frac{\left( (p_t \cdot (u_0 \circ \phi_t^{-1})) * \mathcal{N}(0, \sigma^2) \right)(x)}{(p_t * \mathcal{N}(0, \sigma^2))(x)} \\
&= \frac{(p_t \cdot (u_0 \circ \phi_t^{-1}))(x)}{p_t(x)} \\
&= u_0(\phi_t^{-1}(x)),
\end{aligned}
$$

which proves (24). $\qquad\square$

**Proposition 3.5.** *The marginal vector field $u_t(x)$ defined by (19) and (21) generates the same marginal probability path as the solution $\pi^*$ to the SB problem in (18).*

*Proof of Proposition 3.5.* Using Theorem 2.4 of Léonard (2014a), De Bortoli et al. (2021) showed that the initial and terminal marginals of $\pi^*$ are the solution to the static OT problem

$$\pi^*(x_0, x_1) = \arg\min \text{KL}(\pi^*(x_0, x_1) \| p_{\text{ref}}(x_0, x_1)),$$

while the conditional path distributions $\pi^*(-|x_0, x_1)$ minimize

$$\mathbb{E}_{x_0, x_1 \sim \pi^*(x_0, x_1)} \text{KL}(\pi^*(-|x_0, x_1) \| p_{\text{ref}}(-|x_0, x_1)).$$

The optimization problem for $\pi^*(x_0, x_1)$ is equivalent to the entropy-regularized optimal transport problem with optimum $\pi_{2\sigma^2}$, as observed by De Bortoli et al. (2021). (The key observation is that $\log p_{\text{ref}}(x_0, x_1) = \frac{c(x_0, x_1)^\alpha}{2\sigma^2} + \text{const.}$, where $c(x, y) = \|x - y\|$ and $\alpha = 2$.) The divergences between conditional path distributions are optimized by Brownian bridges with diffusion scale $\sigma$ pinned at $x_0$ and $x_1$, which are well-known to have marginal probability path $p_t$ in (20), and, by (5), are generated by the vector fields $u_t$ in (21). $\qquad\square$

# B    Additional theoretical results

**Proposition B.1.** *For any $\sigma \in \mathbb{R}_+$ conditional flow matching with conditional probability paths given by (16) has an equivalent marginal probability flow $p_t(x)$ to Lipman et al. (2023)'s flow matching.*

*Proof.* To prove the proposition, we use the fact that the Gaussian family can be generated by location-scale transformations (see, *e.g.*, Lehmann & Casella, 2006), *i.e.*, we can express any Gaussian $Z_0 \sim \mathcal{N}(\mu_0, \sigma_0^2)$ as $Z_0 = \mu_0 + \sigma_0 Z$ where $Z \sim \mathcal{N}(0, \boldsymbol{I})$. Recall that the density $p_t(x)$ has the form $p_t(x) = \int p_t(x|z)q(z)dz$, to show the equivalence between the flow from FM and source conditional flow matching, we have to show that $p_t(x|x_1)$ is the same for both methods, that is we show that CFM with variance $(\sigma t)^2 + 2\sigma t(1-t)$ is equivalent to FM with variance $(t\sigma - t + 1)^2$ (12). Since $p_t(x|x_0, x_1)$ is $\mathcal{N}(tx_1 + (1-t)x_0), \sigma t(\sigma t - 2t + 2))$ we can write the random variable $X|X_0, X_1$ as

$$X|X_0, X_1 = tx_1 + (1-t)x_0 + (\sigma t(\sigma t - 2t + 2))^{1/2}Z,$$

where $Z \sim \mathcal{N}(0, \boldsymbol{I})$. Without conditioning on $X_0$, we have

$$X|X_1 = tx_1 + (1-t)X_0 + (\sigma t(\sigma t - 2t + 2))^{1/2}Z.$$

By assumption $X_0 \sim \mathcal{N}(0, \boldsymbol{I})$, thus $X|X_1$ is Gaussian, since a linear transformation of Gaussian distributions is also Gaussian. To define its distribution, we only have to define its expectation and variance. By linearity of expectation, we find $\mathbf{E}(X|X_1) = tx_1$, and by independence of $X_0$ and $Z$ we have

$$\text{Var}(X|X_1) = (1-t)^2\text{Var}(X_0) + (\sigma t(\sigma t - 2t + 2))\text{Var}(Z)$$
$$= (1-t)^2 + (\sigma t(\sigma t - 2t + 2)) = (t\sigma - t + 1)^2,$$

hence the flow from source conditional flow matching is the same as FM.    □

**Proposition B.2.** *If $\pi$ is a Monge map, the objective variance of OT-CFM goes to zero as $\sigma \to 0$, i.e.,*

$$\mathbb{E}_{q(z)}\|u_t(x|z) - u_t(x)\|^2 \to 0 \ \ as \ \sigma \to 0$$

*for $u_t(x|z)$ in (15).*

*Proof.* This follows from a basic fact about the transport plan $\pi$. Specifically, that as $\sigma \to 0$, $D_{\text{KL}}(p_t(x|z^i)\|p_t(x|z^j)) \to \infty$ for an $t, x$ for two distinct $z^i$, $z^j$. This means that $p_t(x|z) = p_t(x)$ for any $t, x, z$ therefore

$$u_t(x) = \mathbb{E}_{q(z)}u_t(x|z)p_t(x|z)/p_t(x)$$
$$= u_t(x|z)$$

□

**Proposition B.3.** *The conditional vector field $u_t(x|\bar{z})$ defined by (26) converges to marginal vector field $u_t(x)$ defined by (9) as $m$ goes to population size, i.e.,*

$$\|u_t(x|\bar{z}) - u_t(x)\|^2 \to 0$$

*as $m \to |\mathcal{X}|$.*

*Proof.* As $|z| \to |\mathcal{X}|$, by definition,

$$u_t(x|\bar{z}) = \frac{\sum_i^m u_t(x|z^i)p_t(x|z^i)q(z^i)}{\sum_i^m p_t(x|z^i)q(z^i)}$$
$$= \frac{\sum_{z \in \mathcal{X}} u_t(x|z)p_t(x|z)q(z)}{\sum_{z \in \mathcal{X}} p_t(x|z)q(z)}$$
$$= \mathbb{E}_{q(z)}\frac{u_t(x|z)p_t(x|z)}{p_t(x)}$$
$$= u_t(x)$$

□

# C   Algorithm extensions

In Alg. 1 we presented the general algorithm for conditional flow matching given $q(z)$, $p_t(x|z)$, $u_t(x|z)$. In Table 1 we presented a number of settings of these leading to interesting probability paths. In practice, we may wish to compute $q(z)$ on the fly. Therefore in Alg. 2, Alg. 3, and Alg. 4, we give algorithms for the simplified conditional flow matching, and minibatch versions of OT conditional flow matching and Schrödinger bridge conditional flow matching. In general these consist of first sampling a batch of data from both the source and the target empirical distributions, then resampling pairs of data either randomly (CFM) or according to some OT plan (OT-CFM and SB-CFM).

## C.1   Variance reduction by averaging across batches

An interesting consequence of introducing optimal transport to conditional flow matching is that it greatly reduces variance of the regression target. Informally, as $\sigma \to 0$, $\mathbb{E}_{x,t,z}\|u_t(x|z) - u_t(x)\|^2 \to 0$ for OT-CFM and SB-CFM, which is not true of previous probability paths in Table 1 (See Proposition B.2 for a precise statement). As flow models get larger, more powerful, and more costly, reducing objective variance, and thereby faster training may lead to significant cost savings (Watson et al., 2022b). To this end we also explore reducing the variance of the objective by averaging over a batch. This is not feasible in score matching where the flow conditioned on multiple datapoints is complex. Our CFM framework naturally extends from a pair of datapoints to a batch of pairs. Instead of conditioning on a single pair of datapoints we can condition on a batch of pairs. As the batch increases in size, we trade higher cost in computing the target for lower variance in the target as the batch size increases, the variance in the target goes to zero (see Proposition B.3 for a precise statement).

As formalized in Proposition B.3, we can reduce variance in the target by averaging over multiple datapoints. Specifically, in this case we let $\bar{\boldsymbol{z}} \coloneqq \{z^i \coloneqq (x_0^i, x_1^i)\}_{i=1}^m$, where $z^i$ are i.i.d. from $q(z)$ and

$$p_t(x|\bar{\boldsymbol{z}}) = \frac{\sum_i^m p_t(x|z^i)q(z^i)}{\sum_i^m q(z^i)} \tag{25}$$

$$u_t(x|\bar{\boldsymbol{z}}) = \frac{\sum_i^m u_t(x|z^i)p_t(x|z^i)q(z^i)}{p_t(x|\bar{\boldsymbol{z}})} \tag{26}$$

It takes roughly $m$ times as long to compute the conditional target $u_t(x|\bar{\boldsymbol{z}})$ but reduces the variance. As the evaluation and backpropagation through $v_\theta$ gets more difficult this tradeoff can be beneficial.

## C.2   Modeling energy functions

If we have access to an energy function two (unnormalized) energy functions $\mathcal{R}_{\{0,1\}} : \mathbb{R}^d \to \mathbb{R}^+$ at the endpoints instead of i.i.d. samples $\boldsymbol{X}_t \sim q_t(x_t)$, then the objective must be slightly modified. We formulate the Energy Conditional Flow Matching Objective as

$$\mathcal{L}_{\text{ECFM}} = \mathbb{E}_{t,\hat{q}_0(x_0),\hat{q}_1(x_1),p_t(x|x_0,x_1)} \left[ \frac{R_0(x_0)R_1(x_1)}{\hat{q}_0(x_0)\hat{q}_1(x_1)} \|v_\theta(t,x) - u_t(x|\boldsymbol{x}_0,\boldsymbol{x}_1)\|_2^2 \right] \tag{27}$$

We can use this object to train a flow which matches the energies without access to samples. This is formalized in the following theorem.

**Proposition C.1.** *Assuming that $\hat{q}_{\{0,1\}}(x), p_t(x) > 0$ for all $x \in \mathcal{X}$ and $t \in [0, 1]$ then the gradients of $\mathcal{L}_{\text{FM}}$ and $\mathcal{L}_{\text{ECFM}}$ with respect to $\theta$ are equal up to some multiplicative constant $c$.*

$$\nabla_\theta \mathcal{L}_{\text{FM}}(\theta) = c \nabla_\theta \mathcal{L}_{\text{ECFM}}(\theta) \tag{28}$$

---

**Algorithm 2** Simplified Conditional Flow Matching (I-CFM)

---

**Input:** Empirical or samplable distributions $q_0, q_1$, bandwidth $\sigma$, batchsize $b$, initial network $v_\theta$.
**while** Training **do**

> /* Sample batches of size b i.i.d. from the datasets                               */
> $\boldsymbol{x}_0 \sim q_0(\boldsymbol{x}_0); \quad \boldsymbol{x}_1 \sim q_1(\boldsymbol{x}_1)$
> $t \sim \mathcal{U}(0, 1)$
> $\mu_t \leftarrow t\boldsymbol{x}_1 + (1-t)\boldsymbol{x}_0$
> $x \sim \mathcal{N}(\mu_t, \sigma^2 I)$
> $\mathcal{L}_{\mathrm{CFM}}(\theta) \leftarrow \|v_\theta(t, x) - (\boldsymbol{x}_1 - \boldsymbol{x}_0)\|^2$
> $\theta \leftarrow \mathrm{Update}(\theta, \nabla_\theta \mathcal{L}_{\mathrm{CFM}}(\theta))$

**return** $v_\theta$

---

**Algorithm 3** Minibatch OT Conditional Flow Matching (OT-CFM)

---

**Input:** Empirical or samplable distributions $q_0, q_1$, bandwidth $\sigma$, batch size $b$, initial network $v_\theta$.
**while** Training **do**

> /* Sample batches of size b i.i.d. from the datasets                               */
> $\boldsymbol{x}_0 \sim q_0(\boldsymbol{x}_0); \quad \boldsymbol{x}_1 \sim q_1(\boldsymbol{x}_1)$
> $\pi \leftarrow \mathrm{OT}(\boldsymbol{x}_1, \boldsymbol{x}_0)$
> $(\boldsymbol{x}_0, \boldsymbol{x}_1) \sim \pi$
> $\boldsymbol{t} \sim \mathcal{U}(0, 1)$
> $\mu_t \leftarrow \boldsymbol{t}\boldsymbol{x}_1 + (1-\boldsymbol{t})\boldsymbol{x}_0$
> $\boldsymbol{x} \sim \mathcal{N}(\mu_t, \sigma^2 I)$
> $\mathcal{L}_{\mathrm{CFM}}(\theta) \leftarrow \|v_\theta(\boldsymbol{t}, \boldsymbol{x}) - (\boldsymbol{x}_1 - \boldsymbol{x}_0)\|^2$
> $\theta \leftarrow \mathrm{Update}(\theta, \nabla_\theta \mathcal{L}_{\mathrm{CFM}}(\theta))$

**return** $v_\theta$

---

*Proof.* Let $z_0 = \int_{\mathcal{X}} R_0(x)dx$, and $z_1 = \int_{\mathcal{X}} R_1(x)dx$ then $q(x_0) = \mathcal{R}_0(x_0)/z_0$, similarly $q(x_1) = \mathcal{R}_1(x_1)/z_1$, then

$$\mathcal{L}_{\mathrm{ECFM}}(\theta) = \mathbb{E}_{t,\hat{q}_0(x_0),\hat{q}_1(x_1),p_t(x|x_0,x_1)}\left[\frac{\mathcal{R}_0(x_0)\mathcal{R}_1(x_1)}{\hat{q}_0(x_0)\hat{q}_1(x_1)}\|v_\theta(t,x) - u_t(x|\boldsymbol{x}_0,\boldsymbol{x}_1)\|_2^2\right] \tag{29}$$

$$= z_0 z_1 \mathbb{E}_{t,\hat{q}_0(x_0),\hat{q}_1(x_1),p_t(x|x_0,x_1)}\left[\frac{q_0(x_0)q_0(x_1)}{\hat{q}_0(x_0)\hat{q}_1(x_1)}\|v_\theta(t,x) - u_t(x|\boldsymbol{x}_0,\boldsymbol{x}_1)\|_2^2\right] \tag{30}$$

$$= z_0 z_1 \int_{t,x_0,x_1,x}\left[q_0(x_0)q_1(x_1)\|v_\theta(t,x) - u_t(x|\boldsymbol{x}_0,\boldsymbol{x}_1)\|_2^2\right]p_t(x|x_0,x_1)dx_0 dx_1 dx \tag{31}$$

$$= z_0 z_1 \mathcal{L}_{\mathrm{CFM}}(\theta) \tag{32}$$

where we use substitution for the first step and change the order of integration in the last step. With an application of Theorem 3.2 the gradients are equivalent up to a factor of $z_0 z_1$ which does not depend on $x$. $\qquad\square$

Of course $\mathcal{L}_{\mathrm{ECFM}}$ leaves the question of sampling open for high-dimensional spaces. Sampling uniformly does not scale well to high dimensions, so for practical reasons we may want a different sampling strategy.

We use this objective in Fig. D.9 with a uniform proposal distribution as a toy example of this type of training.

## D   Additional results

We start this section by the definition of the entropy regularized OT problem:

$$W(q_0, q_1)_{2,\lambda}^2 = \inf_{\pi_\lambda \in \Pi} \int_{\mathcal{X}^2} c(x, y)^2 \pi_\lambda(dx, dy) - \lambda H(\pi), \tag{33}$$

---

**Algorithm 4** Minibatch Schrödinger Bridge Conditional Flow Matching (SB-CFM)

---

**Input:** Empirical or samplable distributions $q_0, q_1$, bandwidth $\sigma$, batch size $b$, initial network $v_\theta$.
**while** Training **do**
  /* *Sample batches of size b i.i.d. from the datasets* */
  $\boldsymbol{x}_0 \sim q_0(\boldsymbol{x}_0); \quad \boldsymbol{x}_1 \sim q_1(\boldsymbol{x}_1)$
  $\pi_{2\sigma^2} \leftarrow \text{Sinkhorn}(\boldsymbol{x}_1, \boldsymbol{x}_0, 2\sigma^2)$
  $(\boldsymbol{x}_0, \boldsymbol{x}_1) \sim \pi_{2\sigma^2}$
  $\boldsymbol{t} \sim \mathcal{U}(0, 1)$
  $\mu_t \leftarrow \boldsymbol{t}\boldsymbol{x}_1 + (1-\boldsymbol{t})\boldsymbol{x}_0$
  $\boldsymbol{x} \sim \mathcal{N}(\mu_t, \sigma^2 \boldsymbol{t}(1-\boldsymbol{t})I)$
  $\boldsymbol{u}_t(\boldsymbol{x}|\boldsymbol{z}) \leftarrow \frac{1-2\boldsymbol{t}}{2\boldsymbol{t}(1-\boldsymbol{t})}(\boldsymbol{x} - (\boldsymbol{t}\boldsymbol{x}_1 + (1-\boldsymbol{t})\boldsymbol{x}_0)) + (\boldsymbol{x}_1 - \boldsymbol{x}_0)$                 ▷ *From (21)*
  $\mathcal{L}_{\text{CFM}}(\theta) \leftarrow \|v_\theta(\boldsymbol{t}, \boldsymbol{x}) - \boldsymbol{u}_t(\boldsymbol{x}|\boldsymbol{z})\|^2$
  $\theta \leftarrow \text{Update}(\theta, \nabla_\theta \mathcal{L}_{\text{CFM}}(\theta))$
**return** $v_\theta$

---

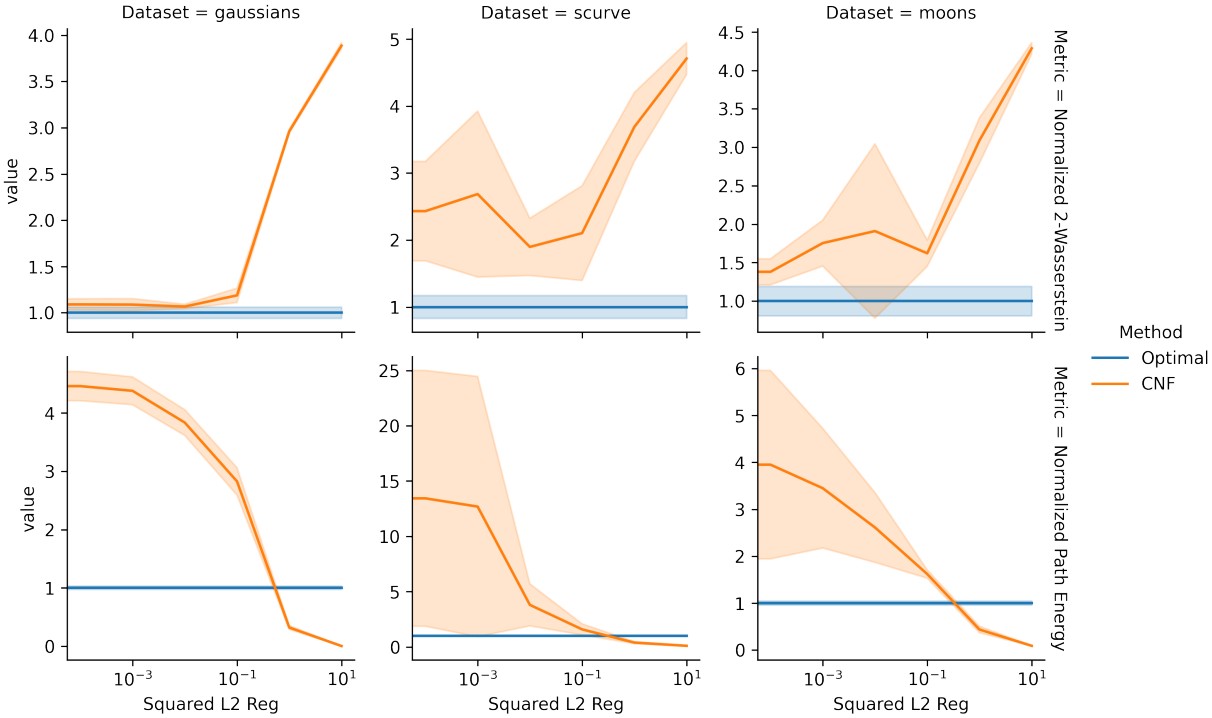

Figure D.1: Evaluation of regularization strength of $\lambda_e$ over 6 seeds in the range $[0, 10^{-5}, 10^2]$. $\lambda_e = 0.1$ performs the best in terms of minimizing path length and test error. We call this model "Regularized CNF".

where $\lambda \in \mathbb{R}^+$ and $H(\pi) = \int \ln \pi(x, y) d\pi(dx, dy)$.

**Regularized CNF tuning** Continuous normalizing flows with a path length penalty optimize a relaxed form of a dynamic optimal transport problem (Tong et al., 2020; Finlay et al., 2020; Onken et al., 2021). Where dynamic optimal transport solves for the optimal vector field in terms of average path length where the marginals at time $t = 0$ and $t = 1$ are constrained to equal two input marginals $q_0$ and $q_1$. Instead of this pair of hard constraints, regularized CNFs instead set $q_0 := \mathcal{N}(x \mid 0, 1)$ and optimize a loss of the form

$$L(x(t)) = -\log p(x(t)) + \lambda_e \int_0^1 \|v_\theta(t, x(t))\|^2 dt \tag{34}$$

Table D.1: Mean training time till convergence in $10^3$ seconds over 5 seeds, with the exception of DSB, trained over 1 seed. CFM variants and DSB are trained on a single CPU with 5GB of memory where other baselines are given two CPUs and one GPU. CFM, with significantly fewer resources, still trains the fastest.

| | $\mathcal{N}\to$8gaussians | moons$\to$8gaussians | $\mathcal{N}\to$moons | $\mathcal{N}\to$scurve | mean |
|---|---|---|---|---|---|
| OT-CFM | $1.284 \pm 0.028$ | $1.587 \pm 0.204$ | $1.464 \pm 0.158$ | $1.499 \pm 0.157$ | $1.484 \pm 0.192$ |
| CFM | $0.993 \pm 0.021$ | $1.102 \pm 0.171$ | $\mathbf{1.059 \pm 0.158}$ | $\mathbf{1.008 \pm 0.106}$ | $1.046 \pm 0.132$ |
| FM | $0.839 \pm 0.096$ | — | $1.076 \pm 0.126$ | $1.127 \pm 0.123$ | $1.014 \pm 0.170$ |
| SB-CFM | $\mathbf{0.713 \pm 0.386}$ | $\mathbf{0.794 \pm 0.293}$ | $1.143 \pm 0.389$ | $1.230 \pm 0.424$ | $\mathbf{0.935 \pm 0.397}$ |
| Reg. CNF | $2.684 \pm 0.052$ | — | $9.154 \pm 1.535$ | $9.022 \pm 3.207$ | $8.021 \pm 3.288$ |
| CNF | $1.512 \pm 0.234$ | — | $17.124 \pm 4.398$ | $27.416 \pm 13.299$ | $18.810 \pm 12.677$ |
| ICNN | $3.712 \pm 0.091$ | $3.046 \pm 0.496$ | $2.558 \pm 0.390$ | $2.200 \pm 0.034$ | $2.912 \pm 0.626$ |
| DSB | $5.418 \pm$ — | $5.682 \pm$ — | $5.428 \pm$ — | $5.560 \pm$ — | $5.522 \pm$ — |

where $\frac{dx}{dt} = v_\theta(t, x(t))$ and $\log p(x(T))$ is defined as

$$\log p(x(T)) = p(x(0)) + \int_0^T \frac{\partial \log p(x(t))}{\partial t} dt = p(x(0)) + \int_0^T -\mathrm{tr}\left(\frac{dv_\theta}{dx(t)}\right) dt \tag{35}$$

where the second equality follows from the instantaneous change of variables theorem (Chen et al., 2018, Theorem 1). In practice it is difficult to pick a $\lambda_e$ which both produces flows with short paths and allows the model to fit the data well. We analyze the effect of this parameter over three datasets in Fig. D.1. In this figure we analyze the Normalized 2-Wasserstein to the target distribution (which approaches 1 with good fit), and the Normalized Path Energy (NPE). We find a tradeoff between short paths (Low NPE) and good fit (Low 2-Wasserstein). We choose $\lambda_e = 0.1$ as a good tradeoff across datasets, which has paths that are not too much longer than optimal but also fits the data well.

**Ablation results on batch size.** Since we use Minibatch-OT for OT-CFM, when the minibatch size is equal to one, then OT-CFM is equivalent to CFM. This effect can be seen in Fig. D.2, where over four datasets, OT-CFM starts with equal path length and approximately equal 2-Wasserstein. Then the normalized path energy decreases surprisingly quickly plateauing after batchsize reaches $\sim$64. While the minibatch size needed to approximate the true dynamic optimal transport paths will vary with dataset (for example in the moon-8gaussian case we need a larger batch size) it is still somewhat surprising that such small batches are needed as this is less than 0.5% of the entire 10k point dataset per batch.

**The effect of $\sigma$ on fit and path length.** Next we consider $\sigma$, the bandwidth parameter of the Gaussian conditional probability path. In Fig. D.3 we study the effect of $\sigma$ on the fit (top) and the path energy (bottom). With $\sigma > 1$ methods start to underfit with high 2-Wasserstein error and either very long or very short paths. As for specific models, SB-CFM becomes unstable with $\sigma$ too small due to the lack of convergence for the static Sinkhorn optimization with small regularization. FM and CFM follow similar trends where they fit fairly well with $\sigma \leq 1$ but have paths that are significantly longer than optimal by 2-3x. OT-CFM maintains near optimal path energies and near optimal fit until $\sigma > 1$.

**Schrödinger bridge fit over simulation time.** In Fig. D.7 we compare the fit of Diffusion Schrödinger Bridge model with SB-CFM conditioned on time. The Diffusion Schrödinger Bridge seems to outperform SB-CFM early in the trajectory, however fails to fit the bridge after many integration steps.

### D.1 Objective variance.

We consider the variance of the objective $u_t(x|z)$ with respect to $z$. While for any $x$ we have $\mathbb{E}_q(z)u_t(x|z) = u_t(x)$, we find a lower second moment speeds up training. Specifically, we seek to understand the effect of the second moment which we call the *objective variance* defined as

$$OV = \mathbb{E}_{t\sim U(0,\mathbf{I}),x\sim p_t(x),z\sim q(z)}\|u_t(x|z) - u_t(x)\|^2 \tag{36}$$

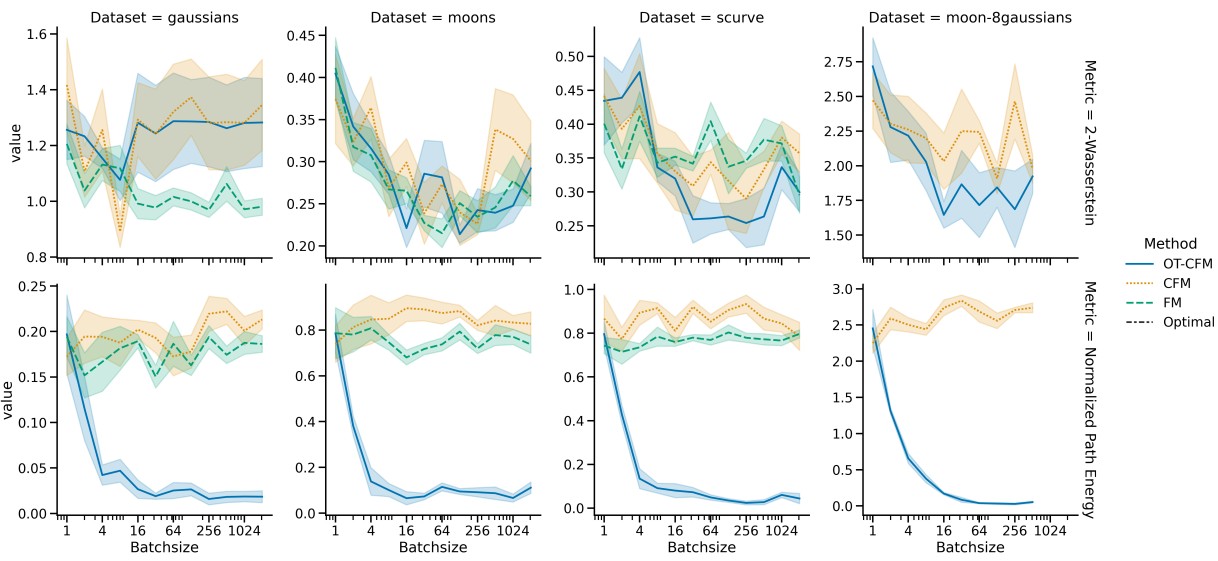

Figure D.2: $\mu \pm \sigma$ of mean path length prediction error over 5 seeds. Lower is better. Introducing OT to CFM batches straightens paths lowering cost towards the optimal $W_2$ as compared to a standard random conditional flow matching network over all batch sizes.

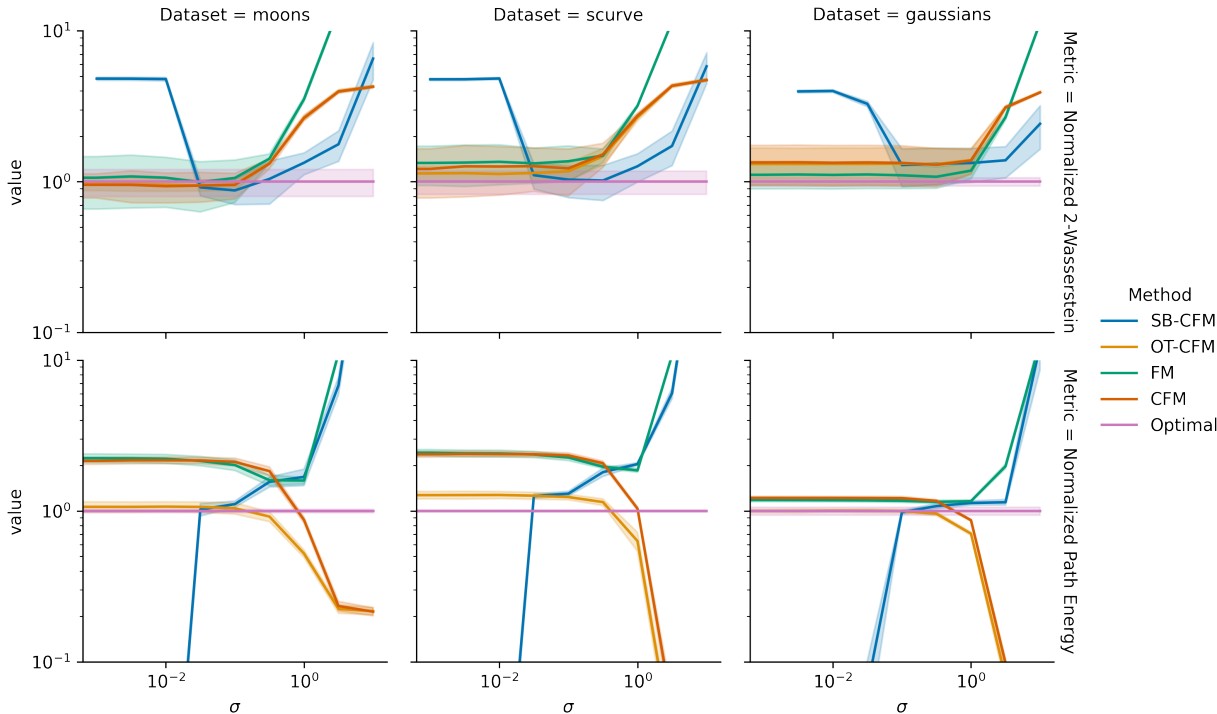

Figure D.3: Evaluation of the effect of $\sigma$ for conditional flow matching models. When $\sigma < 1$ OT-CFM outperforms the other flow matching methods. SB-CFM drops off in performance when $\sigma$ is too small due to numerical issues in the discrete Sinkhorn solver.

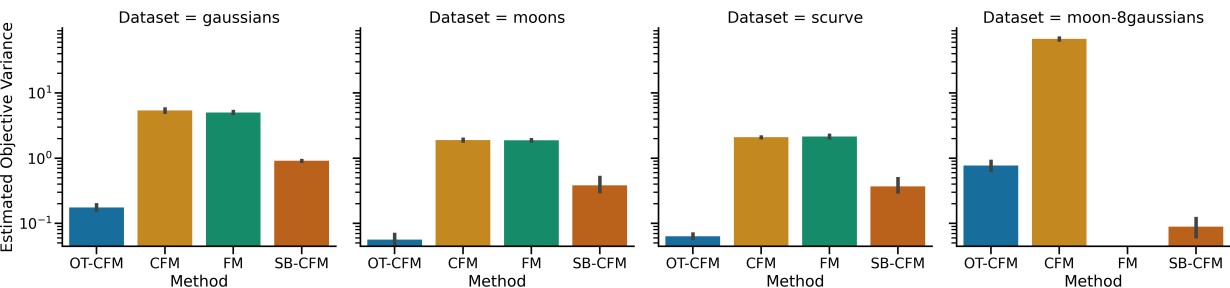

Figure D.4: Estimated Objective Variance (36) for different methods with batch size 512, $\sigma = 0.1$ across datasets. OT-CFM and SB-CFM have significantly lower objective variance than CFM and FM which have roughly equivalent objective variance.

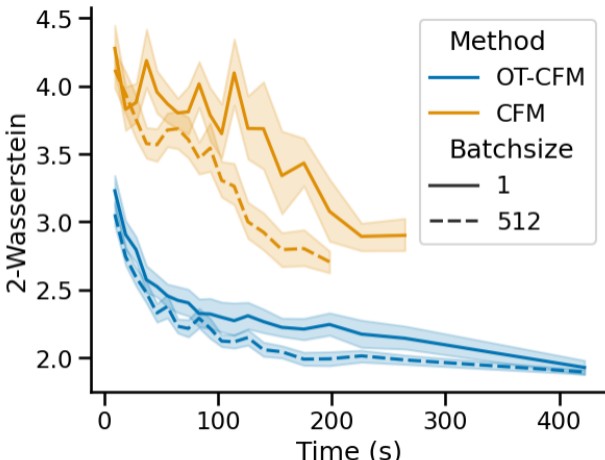

Figure D.5: Validation 2-Wasserstein distance against training time with variance reduction by aggregation either with no aggregation (Batchsize 1) or aggregation over a minibatch (Batchsize 512). Variance reduction leads to faster training, especially for CFM where the objective variance is naturally larger than OT-CFM which sees a small performance gain.

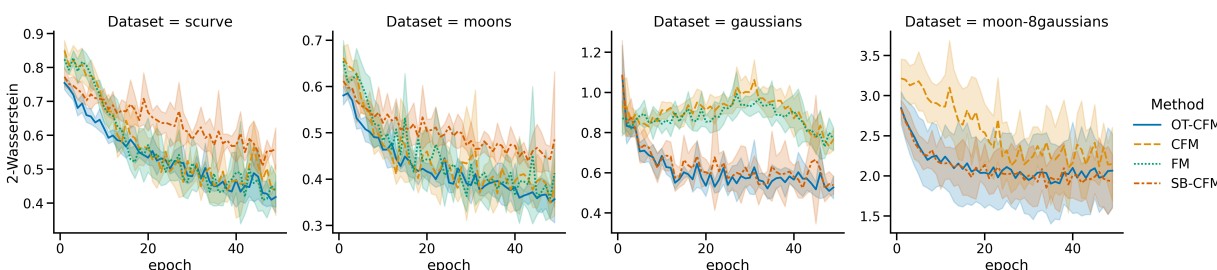

Figure D.6: Extended results from Fig. 2 (left) over two more datasets. OT-CFM is still consistently the fastest converging method.

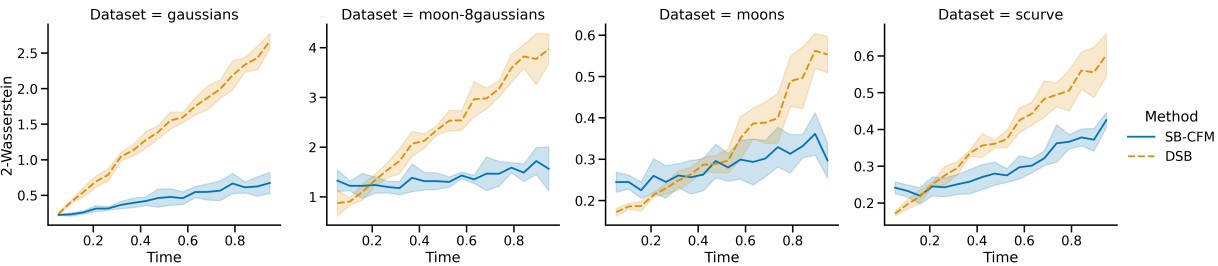

Figure D.7: 2-Wasserstein Error between trajectories and ground truth Schrödinger Bridge samples over simulation time.

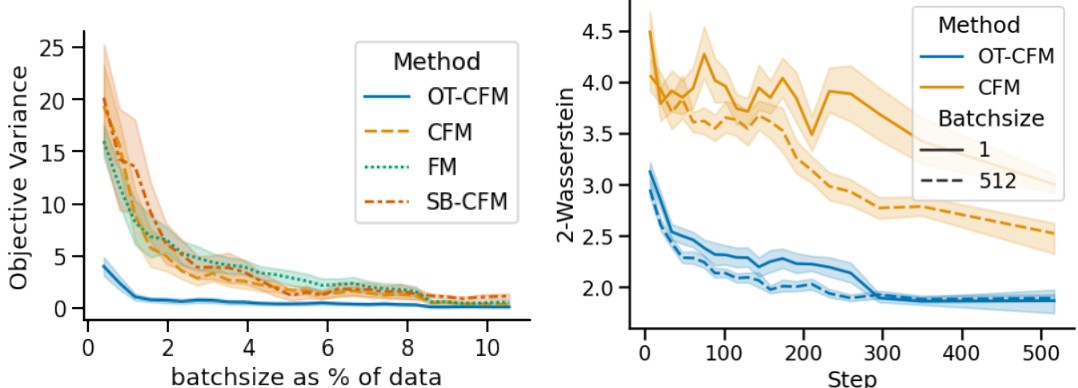

Figure D.8: (left) Variance of the objective for varying batch size. OT-CFM has a lower variance across batch sizes. (right) Validation 2-Wasserstein performance with batch averaging as in §C.1. Reducing variance improves training efficiency.

on training speed for different objectives in Table 1. We estimate the variance on a small data with a known $u_t(x)$. We examine this estimated objective variance and its effect on training convergence in Fig. D.8, showing that either OT-CFM or variance reduced CFM with averaging over the batch results in lower variance of the objective. This in turn leads to faster training times as shown on the right. Averaging over a batch of data leads to faster training particularly for methods with high objective variance (CFM) and less so for those with low (OT-CFM), which already trains quickly.

Variance in the conditional objective target $u_t(x|z)$ varies across models. In Fig. D.4 we study the objective variance across CFM objective functions. Here we estimate the objective variance in (36) as

$$\mathbb{E}_{x,t,z}\|u_t(x|z) - v_\theta(t,x)\|^2 \tag{37}$$

after training has converged. After training has converged $v_\theta$ should be very close to $u_t(x)$ so we use it as an empirical estimator of $u_t(x)$ to compute the variance. We find that across all datasets OT-CFM and SB-CFM have at least an order of magnitude lower variance than CFM and FM objectives. This correlates with faster training as measured by lower validation error in fewer steps for lower variance models as seen in Fig. 2 (left).

We examine the objective variance OV by conditioning $u_t(x|\boldsymbol{z})$ on a batch of pairs of data points, $\bar{\boldsymbol{z}} := \{z^i := (x_0^i, x_1^i)\}_{i=1}^m$, we can reduce the variance of the OV objective to 0 for all models as batchsize goes to population size. For the batchsize $m$ range from 1 to the number of the population, we uniformly sample $m$ pairs of points $z^i$ and compute the probability $p_t(x|\bar{\boldsymbol{z}})$ and the objective $u_t(x|\bar{\boldsymbol{z}})$ from (25) and (26).

We also find that averaging over batches makes the network acheive a lower validation error in fewer steps and in less walltime (Fig. D.5).

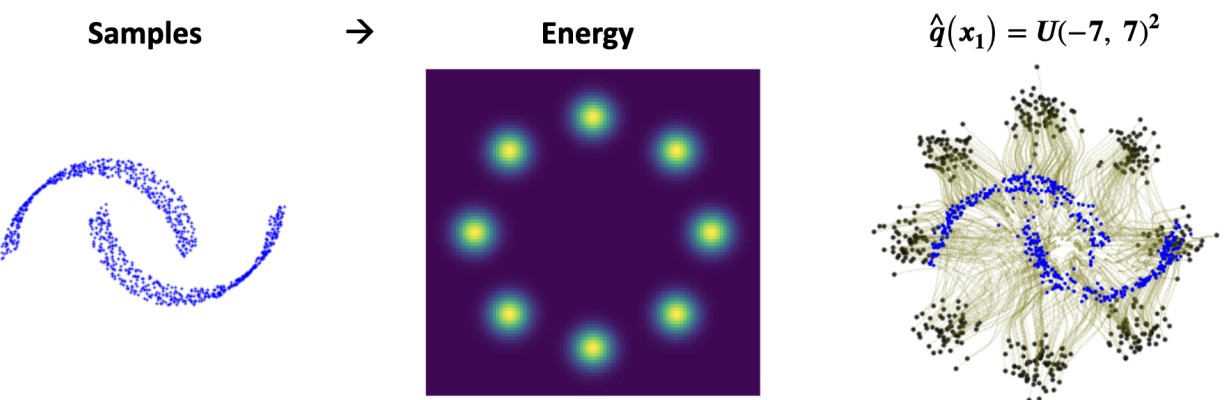

Figure D.9: Flows (green) from (a) moons to (b) 8-Gaussians unnormalized density function learned using CFM with RWIS.

## D.2 Energy-based CFM

We show how CFM and OT-CFM can be adapted to the case where we do not have access to samples from the target distribution, but only an unnormalized density (equivalently, energy function) of the target, $R(x_1)$ (Fig. D.9). We consider the 10-dimensional funnel dataset from Hoffman & Gelman (2011). We aim to learn a flow from the 10-dimensional standard Gaussian to the energy function of the funnel. We consider two algorithms, each of which has certain advantages:

(1) Reweighted importance sampling (RWIS): We construct a *weighted* batch of target points $x_1$ by sampling $x_1 \sim \mathcal{N}(\mathbf{0}, \mathbf{I})$ and assigning it a weight of $R(x_1)/\mathcal{N}(x_1; \mathbf{0}, \mathbf{I})$ normalized to sum to 1 over the batch. The FM and CFM objectives handle weighted samples in a trivial way (by simply using the weights as $q(x_1)$ in Table 1), while OT-CFM treats the weights as target marginals in constructing the OT plan between $x_0$ and $x_1$. We expect RWIS to perform well when batches are large and the proposal and target distributions are sufficiently similar; otherwise, numerical explosion of the importance weights can hinder learning.

(2) MCMC: We use samples from a long-run Metropolis-adjusted Langevin MCMC chain on the target density as approximate target samples. We expect this method to perform well when the MCMC mixes well; otherwise, modes of the target density may be missed.

As an evaluation metric, we use the estimation bias of the log-partition function using a reweighted variational bound, following prior work that studied the problem using SDE modeling (Zhang & Chen, 2022). The computation of this metric for CNFs is given in §E.6.

The results are shown in Table D.2. When an adaptive ODE integrator is used, all algorithms achieve similar results (no pair of mean log-partition function estimates is statistically distinguishable with $p < 0.1$ under a Welch's $t$-test) but OT-CFM is about twice as efficient as CFM and FM. However, with a fixed computation budget for ODE integration, OT-CFM performs significantly better.

# E Experiment and implementation details

## E.1 Physical experimental setup

All experiments were performed on a shared heterogenous high-performance-computing cluster. This cluster is primarily composed of GPU nodes with RTX8000, A100, and V100 Nvidia GPUs. Since the network and nodes are shared, other users may cause high variance in the training times of models. However, we believe that the striking difference between the convergence times in Table D.1 and combined with the CFM training setup with a single CPU and the baseline models trained with two CPUS and a GPU, paints a clear picture as to how efficient CFM training is. Qualitatively, we feel that most CFMs converge quite a bit more rapidly than these metrics would suggest, often converging to a near optimal validation performance in minutes.

Table D.2: Energy-based CFM results on the 10-dimensional funnel dataset: log-partition function estimation bias (mean and standard deviation over 10 runs) and time to generate 6000 samples from the trained ODE. With adaptive integration, OT-CFM requires fewer function evaluations. With a fixed-interval solver, OT-CFM has lower discretization error, leading to a better estimate. PIS baseline is from Zhang & Chen (2022).

| | RWIS | | MCMC | |
|---|---|---|---|---|
| | $\log \hat{Z}$ | $\int$ time | $\log \hat{Z}$ | $\int$ time |
| **adaptive Dormand-Prince (tolerance** $0.01$**) integration** | | | | |
| CFM | $-0.068 \pm 0.041$ | $26.6 \pm 8.4$s | $0.029 \pm 0.037$ | $34.6 \pm 6.0$s |
| OT-CFM | $-0.076 \pm 0.098$ | $\mathbf{13.3} \pm 1.7$s | $0.009 \pm 0.045$ | $\mathbf{12.8} \pm 1.2$s |
| FM | $-0.033 \pm 0.057$ | $26.5 \pm 7.7$s | $0.027 \pm 0.031$ | $30.9 \pm 5.8$s |
| **Euler (**$N = 10$**) integration** | | | | |
| CFM | $0.281 \pm 0.202$ | $4.0 \pm 0.8$s | $0.336 \pm 0.030$ | $3.7 \pm 0.7$s |
| OT-CFM | $-\mathbf{0.039} \pm 0.030$ | $4.2 \pm 0.6$s | $\mathbf{0.146} \pm 0.107$ | $4.1 \pm 0.8$s |
| FM | $0.176 \pm 0.044$ | $4.1 \pm 0.7$s | $0.334 \pm 0.066$ | $3.9 \pm 0.6$s |
| **Euler-Maruyama (**$N = 100$**) integration** | | | | |
| PIS (SDE) | | $-0.018 \pm 0.020$ | | |

## E.2 2D, single-cell, and Schrödinger bridge experimental setup

For all experiments we use the same architecture implemented in PyTorch (Paszke et al., 2019). We concatenate the flattened input $x \in \mathbb{R}^d$ and the time $t$ as the $d + 1$ inputs to a network with three hidden layers of width 64 interspersed with SELU activations (Klambauer et al., 2017) followed by a linear output layer of width $d$. This forms our $v_\theta$ for all experiments. For all 2D and single-cell experiments we train for 1000 epochs and implement early stopping on the validation loss which checks the loss on a validation set every 10 epochs and stops training if there is no improvement for 30 epochs. We also set a time limit of 100 minutes for each CFM model. This is hit almost exclusively for SB-CFM models with small $\sigma$ which are unstable to train due to instabilities and non-convergence of the Sinkhorn (Cuturi, 2013) transport plan optimization. We use the AdamW (Loshchilov & Hutter, 2019) optimizer with weight decay $10^{-5}$ with batchsize 512 by default in 2D experiments and 128 in the single cell datasets. For OT-CFM and SB-CFM we use exact linear programming EMD and Sinkhorn algorithms from the python optimal transport package (Flamary et al., 2021) For evaluation of trajectories unless otherwise noted we use the Runge-Kutta45 (rk4) ODE solver with 101 timesteps from 0 to 1.

## E.3 Variance reduction by averaging

We tackle the exploration of the effects of reducing variance of the target $u_t(x|z)$ from two directions. The first is for small example where we can compute the ground truth $u_t(x)$ quickly, and the second is in the setting of trained models where we can estimate $u_t(x)$ with $v_\theta(t, x)$ after $v_\theta$ has converged.

We first consider the convergence of each flow matching objective (OT-CFM, CFM, FM, SB-CFM) to zero as a function of the batch size relative to the dataset size. This is done by first sampling $t, x, z$ then computing the true objective variance across many samples. This appears in Fig. D.8.

We next consider the effect of averaging over a batch to reduce the variance of the objective in Fig. D.8 (right). Here Batchsize refers to the size of the batch we are averaging over. We aggregate this into a single target so that the model sees a single $d$ dimensional target vector for one sampled $x, t$. This means that we can compare different aggregation sizes fairly.

### E.4 Schrödinger bridge evaluation setup

To evaluate how well Schrödinger Bridge models actually model a Schrödinger Bridge, we constrain ourselves to a small example with 1000 points. We note that the closed-form Schrödinger marginals are known for discrete densities, for Gaussians (Mallasto et al., 2022), and can be constructed for two approximate datasets (Korotin et al., 2021), which present other ways of evaluating Schrödinger bridge performance. For any time $t$ we can sample from the ground truth Schrödinger bridge density $p_t(x)$ as

$$(x_0, x_1) \sim \pi_{2\sigma^2}$$
$$X_t \sim \mathcal{N}(x \mid tx_1 + (1-t)x_0, \sigma t(1-t))$$

We sample trajectories of length 20 from $t = 0$ to $t = 1$ by integrating over time from $t = 0$ to $t = 1$. At each of the 18 intermediate timepoints we compute the 2-Wasserstein distance between a sample of size 1000 from the trajectories at that time and the ground truth $X_t$ as above at that time. We reported the average across the 18 intermediate timepoints in Table 3 and plot the 2-Wasserstein distance over time in Fig. D.7.

**SB-CFM Model** We train SB-CFM with $\sigma = 1$ and batchsize=512 for each of the datasets. We save 1000 trajectories from a test set integrated with the tsit5 solver with atol=rtol=1e-4.

**Diffusion Schrödinger bridge model implementation details** We use the implementation from De Bortoli et al. (2021). Only the networks were changed for a fair comparison with CFM. The forward and backward networks are composed of an MLP with three hidden layers of size 64, with SELU activations in between layers. We used a time and a positional encoders composed of two layers of size 16 and 32 with LeakyReLU activations has inputs to the score network. The architectures are the same for the 2D examples and the single-cell examples (except for the input dimension). During training, we set the variance ($\gamma$ in the author's code) to 0.001 and did 20 steps to discretize the Langevin dynamic. We trained for 10k iterations with 10k particles and batch size of 512, for 20 iterative proportional fitting steps, and a learning rate set to 0.0001. For the interpolation task we used the tenth timepoint from the Langevin dynamic with the backward network trained to go from the distribution at time $t - 1$ to $t + 1$. All trajectories are evaluated from the backward dynamic. We use $\sigma = 1$ and batchsize=512.

### E.5 Single-cell experimental setup

We strove to be consistent with the experimental setup of Tong et al. (2020). For the Embryoid body (EB) data, we use the same processed artifact which contains the first 100 principal components of the data. For our tests we truncate to the first five dimensions, then whiten (subtract mean and divide by standard deviation) each dimension. For the Embryoid body dataset which consists of 5 timepoints collected over 30 days we train separate models leaving out times $1, 2, 3$ in turn. We train a CFM over the full time scale (0-4). During testing we push forward all points $X_{t-1}$ to time $t$ as a distribution to test against.

For the Cite and Multi datasets these are sourced from the Multimodal Single-cell Integration challenge at NeurIPS 2022, a NeurIPS challenge hosted on Kaggle where the task was multi-modal prediction (Burkhardt et al., 2022). In this competition they used this data to investigate the predictability of RNA from chromatin accessibility and protein expression from RNA. Here, we repurpose this data for the task of time series interpolation. Both of these datasets consist of four timepoints from CD34+ hematopoietic stem and progenitor cells (HSPCs) collected on days 2, 3, 4, and 7. For more information and the raw data see the competition site.[5] We preprocess this data slightly to remove patient specific effects by focusing on a single donor (donor 13176), then we again compute the first five principal components and again whiten each dimension to further normalize the data.

### E.6 Energy-based CFM

The 10-dimensional funnel dataset is defined by $\mathbf{x}_0 \sim \mathcal{N}(0, \boldsymbol{I})$, $\mathbf{x}_{1,\dots,9} \sim \mathcal{N}(\mathbf{0}, \exp(\mathbf{x}_0)\mathbf{I})$. We attempted to mimic the SDE model architecture from Zhang & Chen (2022) for the flow model $v_\theta(t, x)$. The time step $t$ is

---

[5]https://www.kaggle.com/competitions/open-problems-multimodal/data

encoded with 128-dimensional Fourier features, then both $x$ and $t$ are independently processed with two-layer MLPs. The two representations are concatenated and processed through another three-layer MLP to make the prediction. All MLPs use GELU activation and have 128 units per hidden layer. We trained all models with $\sigma = 0.05$ and learning rate $10^{-2}$, the highest at which they were table, for 1500 batches of size 300, to be consistent with the settings from Zhang & Chen (2022).

The importance-weighted estimate of the log-partition function is defined

$$\log \hat{Z} = \log \frac{1}{K} \sum_{i=1}^{K} \frac{R(x_1^{(i)})}{\mathcal{N}(x_0^{(i)}; 0, \mathbf{I})} \left| \frac{\partial x_1}{\partial x_0} \right|_{x_0 = x_0^{(i)}},$$

where $x_0^{(i)}$ are independent samples from the source distribution and $x_1^{(i)}$ is $x_0^{(i)}$ pushed forward by the flow (note that the Jacobian can be computed by differentiating the ODE integrator). We used $K = 6000$ samples.

For MCMC, to be consistent with Zhang & Chen (2022), we generated 15000 samples, each of which was seen 30 times in training. We used 1000 steps of Metropolis-adjusted Langevin sampling with $\epsilon$ linearly decaying from 0.1 to 0.

The flow network used to generate Fig. D.9 followed similar settings to those used in §5.1.

### E.7 Unsupervised translation

We trained a vanilla convolutional VAE, with about 7 million parameters in the encoder, on CelebA faces scaled to $128 \times 128$ resolution.

For the flow network $v_\theta(t, x)$, we used a MLP with four hidden layers of 512 units and leaky ReLU activations taking the 129-dimensional concatenation of $x$ and $t$ as input. All models CFM and OT-CFM were trained for 5000 batches of size 256 and the Adam optimizer with learning rate $10^{-3}$. Integration was performed using the Dormand-Prince integrator with tolerance $10^{-3}$. For each attribute, 1000 positive and negative images each were used as a held-out test set.

Fig. E.1 shows some examples of the learned trajectories.

### E.8 Unconditional CIFAR-10 experiments

For the CIFAR-10 experiments we followed the setup as described in Lipman et al. (2023). All methods were trained with the same setup, only differeing in the choice of probability path. Since code has not been released for this work, there are a few parameters which may differ. We summarize the setup here, where the exact parameter choices can be seen in the source code.

We used the Adam optimizer with $\beta_1 = 0.9$, $\beta_2 = 0.999$, $\epsilon = 10^{-8}$, and no weight decay. To reproduce Lipman et al. (2023), we used the UNet architecture from Dhariwal & Nichol (2021) with channels $= 256$, depth $= 2$, channels multiple $= [1, 2, 2, 2]$, heads $= 4$, heads channels $= 64$, attention resolution $= 16$, dropout $= 0.0$, batch size per gpu $= 128$, gpus $= 2$, epochs $= 2000$, maximum learning rate $= 5 \times 10^{-4}$, minimum learning rate $= 0$, with a learning schedule that increases linearly from the minimum to the maximum learning rate over the first 200 epochs, and decays linearly from back to the minimum after that. We use $\sigma = 10^{-6}$. For sampling, we use Euler integration using the torchdyn package and DOPRI5 from the torchdiffeq package. Since the DOPRI5 solver is an adaptive step size solver, it uses a different number of steps for each integration. We use a batch size of 500 for a 100 total batches and average the number of function evaluations (NFE) over batches.

For our improved models we use the same parameters as above except we use channels $= 128$, dropout $= 0.1$, a single A100 GPU, steps=400000, $\sigma = 0$, constant learning rate of $2 \times 10^{-4}$, gradient clipping with norm $= 1.0$, and exponential moving average weights with decay $= 0.9999$.

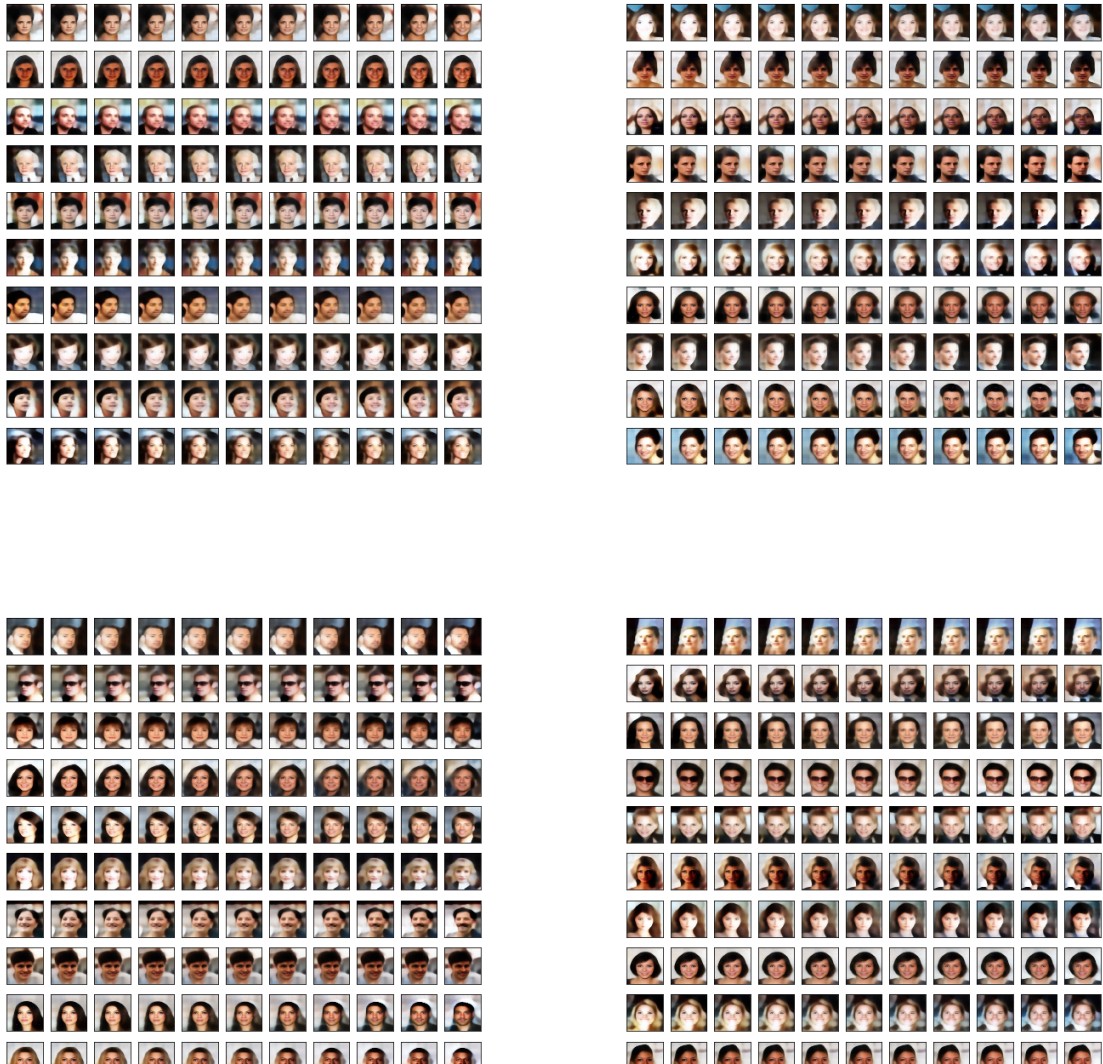

Figure E.1: Image-to-image translation in the latent space of CelebA images: An OT-CNF is trained to translate between latent encodings of images that are negative and positive for a given attribute. The first column is a reconstructed encoding $x_0$ of a real negative image. The next ten columns are decodings of images along the flow trajectory with initial condition $x_0$, with $x_1$ shown in the right column. **Top row:** not smiling $\rightarrow$ smiling, not male $\rightarrow$ male, showing the preservation of image structure and other attributes. **Bottom row:** no mustache $\rightarrow$ mustache, not wearing necktie $\rightarrow$ wearing necktie, showing partial failure modes. Both features are well-predicted by the latent vector, but infrequent in the dataset and highly correlated with other attributes, such as 'male', leading to unpredictable behaviour for out-of-distribution samples and modification of attributes different from the target.

