# OpenReview forum: "Improving and generalizing flow-based generative models with minibatch optimal transport"
_TMLR — Accepted by TMLR_

### Review · Reviewer_gmjr · 2023-11-27

**Summary Of Contributions:**

- The authors propose flow / bridge matching on minibatch OT couplings for regularized and non regularized OT on minibatches
- The authors extend this where the marginals are only known in energy form
- The authors provide code package for their method (though not baselines)
- The authors improve state of the art performance for flow based generative models on cifar10

**Audience:**

Yes

**Claims And Evidence:**

No

**Requested Changes:**

- Please temper the claims stated above. I understand this is one of the earlier works performing bridge / flow matching for (approximate) OT couplings. This has been found to be useful, though I imagine it has not yet been published due to such overreaching claims.
- Provide justification of convergence for flow matching on (changing) minibatch couplings / samples from couplings.
- Please correct lack of citation to prior works such as Peluchetti 2021
- Given it is almost December 2023, I would expect work from August 2023 and earlier to be discussed and cited. Even to state such work is concurrent at the first time of attempted publication.

**Strengths And Weaknesses:**

**Strengths**
- This is one of the early works to use static OT maps (or approximations more precisely) to improve bridge / flow matching procedures. Although the disintegration of KL into coupling and bridge is widely known, it has not been applied before in the context of generative modelling due to the difficulty in computing the coupling.
- The authors show that a minibatch OT heuristic can benefit flow matching training by improving stability and lowering variance of the loss objective
- The authors improve generative performance of flow matching for cifar10 benchmark to around fid 3.6. Although this does not compete with diffusion, diffusion bridge or IPF based SB [7] methods at around FID 1.7/ 1.7 / 3.0 respectively, it is an improvement over prior flow based approaches.
- The torchCFM package is a nice addition (though baselines in the paper appear to be missing)
- The energy based approach appears novel and interesting; extending the methods beyond sample only paradigms


**Weaknesses**
- Some claims are overstated and unsubstantiated
  > OT-CFM is the first method to compute dynamic OT in a simulation-free way

  > Our work is the first to enable simulation-free training of dynamic OT maps and Schrödinger bridge probability flows for arbitrary source and target distributions.

This work relies on minibatch OT, hence reliant on minibatch solvers. It is well known that minibatch solvers output a biased coupling for the true OT between marginal distributions; only coinciding for discrete measures on full batch OT solvers; hence it has not been proven that learning the vector field wrt minibatch couplings will return the SB for *arbitrary source and target distributions* only for discrete measures, even under the perfect training regime in terms of network and training time (not infinite batch size) - this is quite different to other approaches such as IPF / IMF [7,8,11,10] which should converge to (regularized) OT in the perfect training regime with finite batch size.

The proposed method may surpass [7,8,11,10] in terms of practical application for generative modelling where one does not care about OT, however for OT it is not clear that even practically the discrete solvers would be feasible for discrete measures with a large number of points nor if OT solvers for discrete measures approximate OT to any accuracy for continuous measures in high dimension. The sample complexity of the convergence of minibatch OT to full batch is exponentially poor wrt the dimension of the support. Minibatch OT suffers from $\mathcal{O}(n^3)$ complexity for linear solvers in the non regularized case and $\mathcal{O}(n^2)$ complexity in the regularized case with Sinkhorn, hence it is not practical to have high number of samples. It is far from clear how this method would be useful for computing OT beyond low dimensional toy examples other than as some sort of heuristic. Indeed I imagine in the worst case the regularization would need to be high in order to solve resulting in close to independent coupling or Sinkhorn would not converge and hence close to independent coupling.

Furthermore, if one can perform Sinkhorn / minibatch OT and pretend this is the optimal coupling for the marginal measures, then one can simply interpolate between the measures using a Brownian bridge or linear interpolation to return the dynamic coupling. It is not clear why this method would be useful for approximating OT beyond Sinkhorn - which has been applied to point clouds > 10,000 points, more than the number of points considered in the examples of this paper - "we constrain ourselves to a small example with 1000 points".

I do appreciate that OT may not be the goal of this paper, so this should be clarified and focus more on generative modelling. The generative modelling experiments do not reach the performance of many other recent methods. Indeed given glow matching from Gaussian to data is a particular case of diffusion models, I would expect better performance if the emphasis is that this improves flow / diffusion models for specifically generative modelling.

>  CFM is able to learn conditional generative models from any samplable source distribution by conditioning on paired source and target samples, generalizing existing methods (Lipman et al., 2023; Albergo & Vanden-Eijnden, 2023; Liu, 2022; Pooladian et al., 2023)

Liu et al 2022 [9], also use non Gaussian source and target measures in experiments, see Figure 1. Albergo & Vanden-Eijnden, 2023 also use non Gaussian source and target measures, see section 3.1. In addition, earlier work performs bridge matching for non-Gaussian source / targets [5,6,11].

- Convergence guarantees \
The results are presented as if there was a static coupling however given minibatch couplings, it is not clear if any convergence results hold - I imagine they do but as far as I am aware this has not been shown . I am unaware of any proof that suggests the learnt vector field will even converge or the coupling will converge to a fixed distribution, only that the potentials of minibatch OT converges to an invariant measure.

- Lack of discussion to prior work \
The method to perform bridge matching between coupling is well known for a number of years now. As far as I am aware the first notion of forward bridge matching is from Peluchetti Sept 2021[1]. This proves a diffusion process may be approximated between arbitrary measures by learning the drift of the forward process with h-transform from the diffusion bridge between points sampled from a coupling - hence mixture of diffusion bridge. This has been built upon by many works including Liu et al 2022 [2,3] and has a special case for zero noise - known as flow matching [4]. These works are for the diffusion bridge case which can be particularised to flow matching for zero noise. The results for these works are for arbitrary target / source measures though experimentally are only tried on Gaussian source measures.

Just this year a number of published papers perform bridge matching on couplings which again result in SB if the coupling is optimal [5,6] between arbitrary measures. It is an open question of how to compute an optimal coupling in the general case for arbitrary measures (high dimensional, continuous support) and indeed the coupling is never optimal in prior work nor in the work under review here. Given Pooladian 2023 is cited and discussed I would expect these papers preceding it by a couple of months to also be discussed.

- Concerns about baselines \
> We use the implementation from De Bortoli et al. (2021)

I understand this code base does not use continuous time training but discrete; in addition has a different reference diffusion, integration time and hence different regularization for the coupling and does not coincide with Brownian bridge used as ground truth. Was this taken into account during the baselines?

Similarly it is stated 20 diffusion steps were used for De Bortoli et al. (2021) yet for SB-CFM "tsit5 solver with atol=rtol=1e-4" was used. Is this really comparable?

Is the code available for the baselines?


- [1] Peluchetti 2021 https://openreview.net/forum?id=oVfIKuhqfC&referrer=%5Bthe%20profile%20of%20Stefano%20Peluchetti%5D
- [2,3] Liu et al 2022 https://arxiv.org/abs/2209.01170; https://arxiv.org/abs/2208.14699
- [4] Lipman et al 2022 https://arxiv.org/abs/2210.02747
- [5] Somnath et al 2023 https://arxiv.org/abs/2302.11419
- [6] Liu et al 2023 https://arxiv.org/abs/2302.05872
- [7] Chen et al 2022 Likelihood training of SB https://arxiv.org/abs/2110.11291
- [8] Liu 2022 https://arxiv.org/abs/2209.14577
- [9] Liu et al 2022 Flow Straight and Fast: Learning to Generate and Transfer Data with Rectified Flow
- [10] Bortoli et al 2021, https://arxiv.org/abs/2106.01357
- [11] Shi et al 2023, https://arxiv.org/abs/2303.16852

---

> ### Author Response · Authors · 2024-01-13
>
> > Liu et al 2022 [9], also use non Gaussian source and target measures in experiments, see Figure 1. Albergo & Vanden-Eijnden, 2023 also use non Gaussian source and target measures, see section 3.1
>
> Thank you for pointing this out we have clarified this section in the text.
>
> > Please temper the claims stated above.
>
> Thank you for this note. We meant to refer to the small discrete data setting where we can compute the full OT plan, but can see how this can be misleading and therefore we have adapted the claim to say. “We show that in the case where the true transport plan is samplable, our methods approximate the dynamic OT maps and Schrödinger bridge probability flows for arbitrary source and target distributions with simulation-free training.”
>
> > Provide justification of convergence for flow matching on (changing) minibatch couplings / samples from couplings.
>
> We showed that as long as the marginals of the couplings are equal to the initial and terminal data distributions, it does not matter which coupling is used for CFM, hence even independent couplings (minibatch size 1) lead to correct marginals, as do arbitrary couplings (e.g., OT or entropic OT) between pairs of minibatches sampled at uniform from the data.
>
> At intermediate time points, because the objective is a least-squares regression with a stochastic target, it has a unique optimum, equal to the expectation of this target. Therefore, as the loss decreases to its minimum (equal to the variance of the stochastic target), the learned vector field approaches this expectation. As we show, this expectation is the drift of an ODE that transforms the initial distribution to the terminal one.
>
> We leave for future investigation any bounds relating the optimization gap to the discrepancy between the generated and target distributions.
>
>
> > Please correct lack of citation to prior works such as Peluchetti 2021
>
> Thank you, we have updated the manuscript and discussed the provided references.
>
> > I understand this code base does not use continuous time training but discrete; in addition has a different reference diffusion, integration time and hence different regularization for the coupling and does not coincide with Brownian bridge used as ground truth. Was this taken into account during the baselines?
>
> We matched the regularization for the coupling for our baselines (See E.4), but agree that continuous time vs. discrete time training are different. We use continuous-time training for our methods throughout. DSB only applies to the discrete time setting and scales poorly with a large number of timesteps. We believe this is a key advantage of matching-based training.
>
> > Similarly it is stated 20 diffusion steps were used for De Bortoli et al. (2021) yet for SB-CFM "tsit5 solver with atol=rtol=1e-4" was used. Is this really comparable?
>
> We believe our use of more advanced solvers and continuous time is an advantage of the SB-CFM framework, which can use longer trajectories and adaptive step size solvers where DSB cannot.
>
> > Is the code available for the baselines?
>
> Yes. Code for all baselines is available either in the torchcfm package (ICNN, CNF) attached to this submission, or in the original package (DSB), which is also publicly available.

---

> > ### Comment · Reviewer_gmjr · 2024-01-13
> >
> > Thank you for your response.
> >
> > I understand better that any coupling will result in convergence, I guess the coupling used here is not really from OT but a sort of coupling on the augmented state / hierarchical state of the indices of the minibatches and the coupling between minibatches. I have doubts over this being close to the OT coupling or entropic however. This is not explained.
> >
> > > We matched the regularization for the coupling for our baselines (See E.4), but agree that continuous time vs. discrete time training are different. We use continuous-time training for our methods throughout. DSB only applies to the discrete time setting and scales poorly with a large number of timesteps. We believe this is a key advantage of matching-based training.
> >
> > My concern is that the DSB paper uses an Ornstein Uhlenbeck reference diffusion and you are using a Brownian motion reference as ground truth, hence they are not directly comparable unless making some adjustments. Without providing the code or explanation it is not clear if these adjustments were actually made.
> >
> > > We believe our use of more advanced solvers and continuous time is an advantage of the SB-CFM framework, which can use longer trajectories and adaptive step size solvers where DSB cannot.
> >
> > Cannot any diffusion including one from SB be converted into a flow using the probability flow ODE and hence use these solvers?
> >
> > A follow up paper on SB with diffusion models from Chen et al 2022 (https://arxiv.org/abs/2110.11291) uses continuous time training and have a better performing implementation than the original Bortoli et al paper, even though the method is the same. That may be a better baseline if the objective is generative modelling.
> >
> > Overall I think this paper should be accepted. It is interesting and is appealing to the community. I have concerns over some misleading claims over if this minibatch method really solves OT and some baselines are unclear, as discussed above, however for applied practitioners it may be useful.

---

> ### Author Response · Authors · 2024-01-18
>
> Thank you for following up.
>
> >I guess the coupling used here is not really from OT but a sort of coupling on the augmented state / hierarchical state of the indices of the minibatches and the coupling between minibatches. I have doubts over this being close to the OT coupling or entropic however.
>
> The reviewer is correct that in expectation our plan corresponds to the minibatch OT plan. OT-CFM uses an OT plan computed between a source and a target minibatch. In practice, we rely on an incomplete OT plan estimator with one pair of source and target minibatch couple (see Eq.7 in [1] with k=1). This construction is equivalent to drawing samples from the minibatch OT plan **in expectation**. Indeed, [1] showed in their Theorem 2, that the incomplete estimator converges linearly to the complete minibatch OT plan as the number of source and target minibatch couples grows. When entropic OT is used at the minibatch level (as in SB-CFM), we compute the minibatch entropic OT plan in expectation as well. Regarding the number of minibatch couples, it is common in deep learning to use only one pair to reach state-of-the-art performance [2,3]. We will explain this construction with more details in the minibatch OT paragraph. Regarding your question of how close minibatch OT is to OT, this is an open question that has not been answered and is out of the scope of this work. Nevertheless, we note that the distance between minibatch OT and OT as a cost was studied in [4].
>
> However, we would like to point out that when the true OT plan is computable (both numerically and in terms of memory), we can use it within OT-CFM and we would solve dynamic OT. This is of course not possible in a context of large-scale datasets such as image generation and in this case, using minibatch OT as a coupling gives better results (see Table 5) than independent coupling.
>
> [1] Learning with minibatch Wasserstein : asymptotic and gradient properties, Fatras et al, AISTATS 2020.
>
> [2] Learning Generative Models with Sinkhorn Divergences, Genevay et al., AISTATS 2018.
>
> [3] DeepJDOT: Deep Joint Distribution Optimal Transport for Unsupervised Domain Adaptation, Damadoran et al, ECCV 2018.
>
> [4] Optimal transport: Fast probabilistic approximation with exact solvers, Sommerfeld et al., JMLR 2019.
>
> >My concern is that the DSB paper uses an Ornstein Uhlenbeck reference diffusion and you are using a Brownian motion reference as ground truth, hence they are not directly comparable unless making some adjustments.
>
> Thank you for the clarification. This is indeed a subtle yet important point. The DSB code does not supply its exact configuration for the Gaussian to Gaussian setting which makes reproducing the exact setup somewhat difficult. However, while DSB uses the OU in the synthetic data experiments, it uses the Brownian motion (which is a special case of OU) as the reference process in the MNIST → MNIST experiment. We took this configuration for our Gaussian to Gaussian experiment. We will clarify this in the text.
>
> >Cannot any diffusion including one from SB be converted into a flow using the probability flow ODE and hence use these solvers?
>
> Yes, any pair of SDEs can be converted into an ODE by averaging the forward SDE drift and negative backward SDE drift. At convergence, when forward and backward SDEs are reverses of each other, this equals the probability flow ODE. However, since the DSB is trained with a fixed discretization, in practice in our experience it performs the best when integrated with this same discretization.
>
> >A follow up paper on SB with diffusion models from Chen et al 2022 uses continuous time training and have a better performing implementation than the original Bortoli et al paper... That may be a better baseline if the objective is generative modelling.
>
> Thank you for the suggestion. Chen et al. presented an excellent approach for improving the original DSB.
> While Chen et al.’s modeling assumes continuous time, the training procedure still relies on a discretization and simulation step, as it trains on stochastic trajectories sampled from the model. In addition, looking carefully at the code in [Chen et al., 2022], the SDE training is preceded by a denoising score matching pretraining stage, which is alluded to in the paper (p.22) and according to the [official code](​​github.com/ghliu/SB-FBSDE/blob/main/configs/default_cifar10_config.py#L29) runs for 200k iterations, accounting for a substantial portion of the training steps. Interestingly, better performance can be obtained simply by continuing this pretraining until convergence as in [Song et al.](https://openreview.net/pdf?id=PxTIG12RRHS), and it is well understood that generation from diffusion models as SDEs gives better results than integration of the equivalent ODEs, with recent work such as [Hu et al.](https://arxiv.org/abs/2311.15744) arguing that the time discretization degrades the DDIM (i.e., the probability flow ODE of the diffusion model) near the noise endpoint.

---

### Review · Reviewer_c8zh · 2023-12-03

**Summary Of Contributions:**

The work proposes a modification to the flow matching algorithm, leading to a coupling which is closer to an optimal transportation coupling.

The flow matching algorithm for arbitrary source and target distributions works by optimizing the loss
$$\mathbb{E}[\|v_\theta(t, x_t) - u_t(x_t | x_0, x_1)\|^2],$$
where $u_t$ is often chosen to be proportional to $x_1 - x_0$. The distribution over $(x_0, x_1)$ is often chosen to be an independent coupling of $q_0$ and $q_1$ for simplicity. As often done so, the objective is optimized by subsampling minibatches from $q(x_0)$ and $q(x_1)$ to approximate the above loss.

The authors propose to calculate an optimal transport or entropic regularized optimal transport coupling for each of the minibatches to intuitively get straighter lines.

The authors suggests an improvement to the flow matching algorithm, with the goal to learn a coupling that is closer to an optimal transportation coupling.

The training part of the flow matching algorithm, when applied to various source and target distributions, works by minimizing the loss
$$\mathbb{E}_{q(x_0, x_1)}[\|v_\theta(t, x_t) - u_t(x_t | x_0, x_1)\|^2],$$
There are different choices for $u_t$, but a very common one is to set it equal to $x_1 - x_0$.
The distribution $q(x_0, x_1)$ is often chosen as independent coupling of $q_0$ and $q_1$. The optimization is implemented by sampling minibatches from $q(x_0)$ and $q(x_1)$.

One key contribution of the authors is their suggestion to compute either an optimal transport or entropically regularized optimal transport coupling for each minibatch. This method is proposed with the intuition that it would lead to more direct, linear paths.

Furthermore, the authors set up a framework in which they can express multiple related flow matching algorithms, by conditioning on a general variable $z$ which will either be only $x_1$ or $(x_0, x_1)$.

**Audience:**

Yes

**Broader Impact Concerns:**

I do not perceive any specific ethical implications of this work that would require a Broader Impact Statement.

**Claims And Evidence:**

No

**Requested Changes:**

The authors state two main contributions, the first being related to unifying existing literature, the other one related to solving the optimal transport problem. I discussed these claims in the Strength's and Weaknesses sections. The Weaknesses stated there need to be addressed.

Specifically, the paper needs to clarify the novelty of the introduced algorithms. Where similar algorithms exist or when theoretical results are minor generalizations of existing findings, such instances should be prominently noted where these algorithms or theoretical aspects are presented in the paper. Additionally, the framing of these contributions should be appropriately toned.

**Strengths And Weaknesses:**

## Strengths
The approach of presenting flow matching algorithms as conditional on a general variable effectively unifies multiple algorithms within the same framework. The work is polished and well written.

## Weaknesses
The novelty of the stated contributions is unclear. I will now discuss the "summary of the contributions" with which the authors open the paper.
1. In the "Summary of contributions," the authors claim to introduce a "novel class of objectives" and prove their correctness. This class is defined by conditioning on an abstract variable $z$ rather than $x_0$ or $(x_0, x_1)$ as done in earlier works. However, it appears that all specific examples of this broad class of objectives suggested by the authors have been previously implemented in other articles. Moreover, the results and proofs of correctness in this section offer only marginal theoretical advances over the work cited in [1], essentially substituting $x_1$ with $z$.

2a. The authors claim they address the dynamic OT or Schrödinger problem, having only access to the static solutions to these problems. They present a training objective for calculating the drift of a dynamic OT or Schrödinger map from a static solution. This involves selecting $q$ as the OT or OT-entropic regularized coupling. Nevertheless, given a static solution to the OT or Schrödinger problem, a closed form solution for the dynamic versions exists - by simply interpolating static couplings with geodesics/straight lines or bridges of the corresponding stochastic process. This is also quite cheap to simulate. For generative modeling, the static coupling suffices. When having access to a static OT/OT-regularized coupling, the rationale for training the drift using a neural network or the targeted real-life applications remain ambiguous.

2b. The authors then propose that an approximation to the OT/OT-regularized maps can be achieved by calculating the OT/regularized OT solution only per-batch during the training. While this is an intriguing concept, it has already been introduced in the work referenced in [2]. The authors acknowledge this in the "Related works" section, noting that [2] introduces "closely related" objectives. However, it seems they are actually identical.

Additionally, the authors declare in their introduction that their first main contribution is the introduction of a unifying conditional flow matching framework. As previously noted in the Strengths section, this is a noteworthy contribution. However, its utility to the reader is lessened due to the authors' inadequate classification of existing literature within this framework.
The authors apply their framework to specific settings of $z$ and $q(z)$.
- Section 3.2.1 effectively presents the objective from [1] as a particular instance of their own objective.
- However, in 3.2.2, their description of the objectives in [3] and [4] as "closely related" is somewhat ambiguous. This leaves the reader uncertain as to whether [3] and [4] actually represent special cases within the framework they have introduced.
- Sections 3.3.3 and 3.3.4 overlook the fact that the objective they propose has already been introduced in [2]. This acknowledgment is made only subsequently in the Related Works section.


## Minor
1. The $U(0, I)$ should be a $U(0, 1)$ in equation $(3)$.

[1] Flow Matching for Generative Modeling, Yaron Lipman, Ricky T. Q. Chen, Heli Ben-Hamu, Maximilian Nickel, Matthew Le

[2] Multisample Flow Matching: Straightening Flows with Minibatch Couplings, Aram-Alexandre Pooladian, Heli Ben-Hamu, Carles Domingo-Enrich, Brandon Amos, Yaron Lipman, Ricky T. Q. Chen

[3] Michael S. Albergo and Eric Vanden-Eijnden. Building normalizing flows with stochastic interpolants

[4] Qiang Liu. Rectified flow: A marginal preserving approach to optimal transport. arXiv preprint 2209.14577,
2022

---

> ### Author Response · Authors · 2024-01-13
>
> Thank you for your feedback and the time taken to review our manuscript. Below, we answer each of your questions and remarks:
>
> >  it appears that all specific examples of this broad class of objectives suggested by the authors have been previously implemented in other articles.
>
> We note that two examples of the CFM example that we investigate have not been investigated in prior work (OT-CFM and SB-CFM), although OT-CFM was investigated concurrently in [Pooladian et al. 2023].
>
> We have revisited our claim as follows: “We introduce a generalized formulation of the recent *conditional flow matching* framework”.
>
> > When having access to a static OT/OT-regularized coupling, the rationale for training the drift using a neural network or the targeted real-life applications remain ambiguous.
>
> We present two real-life applications where access to the OT-regularized coupling is useful. (1) generative modeling of images and (2) modeling of time series in single-cell, where our methods have shown to improve state-of-the-art methods. We show improved performance on both of these tasks with access to the regularized coupling at little extra cost.
>
> > However, in 3.2.2, their description of the objectives in [3] and [4] as "closely related" is somewhat ambiguous. This leaves the reader uncertain as to whether [3] and [4] actually represent special cases within the framework they have introduced.
>
> Thank you for this note, we have clarified this in section 3.2.2 and added two lines to table 1.

---

### Review · Reviewer_HJwq · 2023-12-11

**Summary Of Contributions:**

Flow matching models are recent generative models that allow faster and more stable training of continuous normalizing flows, as they do not require simulating the SDE/ODE and the backpropagation through the trajectories.
This paper presents a generalization of this framework. Indeed, it shows that it is possible to train flow matching models when only conditional probability paths $p_t(x|z)$, vector fields $u_t(x|z)$ and conditioning distribution $q(z)$ are given. The authors then consider the three relevant applications:
- An independent coupling of the source distributions: in this case the conditional path is $p_t(x_t|x_0, x_1)$ and $q(z) = q_0(x_0) q_1(x_1)$. The authors propose one example of conditional probability paths and vector fields and their methods allows learning the unconditional vector field $u_t(x_t)$, which is intractable otherwise.
- Optimal transport: here the condition distribution is the optimal coupling $\pi$ between the two sources. The authors then notice that the solution of dynamic OT fits exactly their framework; the marginals are $p_t(x_t) = \int p_t(x_t | x_0, x_1) \pi(dx_0, dx_1)$ where $p_t( \cdot | x_0, x_1)$ is known as well as the conditional vector field. The optimal coupling is not known in practice but the authors propose to use minibatch OT to draw samples from the coupling, during training. This is an interesting contribution.
- Schrodinger bridge problem: Similarly, the authors also note that the marginals $p_t(x_t)$ of the solution to the SB problem are mixture of bridges over the solution to the entropy-regularized OT problem, and the bridges aswell as the vector fields are known. Sampling from the coupling is done using the Sinkhorn algorithm.

**Audience:**

Yes

**Claims And Evidence:**

Yes

**Requested Changes:**

The authors should add the appropriate references next to Propositions 3.4 and 3.5 and discuss in more detail the difference with [2]. The authors should also mention that Theorem 1 is a straightforward extension of the first theorem in Lipman, 2023.

**Strengths And Weaknesses:**

Strengths:
- The contributions in this paper are original and the experiments are rather extensive.
- I found the paper to be well written and everything is explained quite well also.

Weaknesses.

The weaknesses/questions I see are rather on the practical side:
- It seems to me that SB-CFM uses the Sinkhorn algorithm in order to sample from the coupling during training whereas DSB tries to learn the coupling. In Table 3 the authors outperform DSB on the 2d example and in Table 4 for the higher dimensional, SB-CFM underperforms wrt DSB. Can the authors explain why this is the case? Can the authors also explain when they expect SB-CFM to perform better than DSB and why?
- it seems that one of the selling points of dynamic OT for generative modeling is that it allows straighter flows than the traditional diffusion models and hence, one can use fewer steps to discretize the ODE, yielding faster inference. In table 5 the authors perform 100 steps for simulation but this is rather large in comparison to what is currently done with denoising diffusion models. For example, with fewer steps (see [1] for example, although it is quite old and far from state of the art), 50 steps yield good enough performance already. It would have been great to include an experiment in which the authors compare OT-CFM with std Gaussian source with traditional diffusion models and compare the quality of the samples with the same number of steps. Even assuming that somehow OT-CFM allows faster sampling than diffusion models, why is it interesting (for generative modeling) to integrate the ODE if one has already access to the static coupling.
- I believe that the paper does not accurately reflect the literature. I think that propositions 3.4 and 3.5 are well-known results but the authors do not give any references. Also, the difference with [2], which is very close to this work, is not clear.

[1] Song, Jiaming, Chenlin Meng, and Stefano Ermon. "Denoising diffusion implicit models."
[2] Aram-Alexandre Pooladian, Heli Ben-Hamu, Carles Domingo-Enrich, Brandon Amos, Yaron Lipman, and Ricky T.Q. Chen. Multisample flow matching: Straightening flows with minibatch couplings.

---

> ### Author Response · Authors · 2024-01-13
>
> Thank you for your feedback and the time taken to review our manuscript. Below, we answer each of your questions and remarks:
>
> > In Table 3 the authors outperform DSB on the 2d example and in Table 4 for the higher dimensional, SB-CFM underperforms wrt DSB. Can the authors explain why this is the case?
>
> For the data in Table 4, a better Schrödinger bridge does not necessarily correlate with better performance on the task, as the task is asking for the model to extrapolate out of distribution on how observed cells evolve. We believe this suggests that the Schrödinger bridge with $\sigma=1$ (our training objective) does not match biology very well as say the optimal transport interpolant ($\sigma=0$) or even random pairing. It is possible that the minibatch coupling is worse than the full batch coupling, however we believe this is unlikely as the minibatch size is ~1/10 of the full batch size. We tried the SB-CFM as we were not sure this was the case a-priori.
> > Can the authors also explain when they expect SB-CFM to perform better than DSB and why?
>
> We expect SB-CFM to perform better on generative modeling tasks particularly in
> * Speed of training (due to simulation-free nature)
> * Speed of inference (due to using an ODE)
> * To demonstrate this we add an experiment with SB-CFM on CIFAR-10. `We get comparable results with SB-CFM (3.7 FID) to an improved version of DSB (SB-FBSDE) which achieves an FID of 3.01 on the CIFAR-10 generation task by fine tuning a pre-trained diffusion model ((NCSN++ which obtains FID 2.38 on its own). We are unaware of a simulation-based Schrödinger bridge approach which obtains competitive results without pretraining on CIFAR-10. We note that our model is much faster and simpler to train (16 A100 hours), and with a much lower memory footprint (no need for caching trajectories), and only requires a single rather than 2 models.
>
> > It would have been great to include an experiment in which the authors compare OT-CFM with std Gaussian source with traditional diffusion models and compare the quality of the samples with the same number of steps.
>
> We thank the reviewer for their suggestion. In the revision we include the result from [Lipman et al. 2023] for a direct comparison of diffusion to flow matching (although not in a competitive setting). In their setting they find an FID of 7.48 for DDPM and a FID of 6.35 for FM. Our goal is not to show flow matching achieves state of the art performance on image generation tasks in absolute terms. Our goal was to show that augmenting the original Flow Matching work with ideas from optimal transport can improve performance, and that it can approximate dynamic OT in some settings.
>
> > why is it interesting (for generative modeling) to integrate the ODE if one has already access to the static coupling.
>
> In settings where we have access to a static coupling between training data, we would often like to extend this coupling to new test data. This is the case in single-cell rna sequencing data where we have an initial collection and would like to extend the model to new settings.
>
> In the unconditional image setting where it is computationally infeasible to calculate the static coupling exactly, we show that learning with a minibatch coupling improves over a random coupling improves generative modeling performance.

---

### Author Response · Authors · 2024-01-13
**Overall Response**

We would like to thank all reviewers for their time and valuable feedback when reviewing our paper. We appreciate their constructive criticisms that aided us in formulating our responses which will serve to improve the overall quality of our paper. We are especially encouraged that the reviewers found our work clear and well written (HJwq, c8zh) We now respond below to the main clarification points below.

All reviewers asked about the utility of approximating the dynamic OT by a neural network when the discrete static OT plan between training samples is available. We discuss this here in the setting of generative modeling.

While the (discrete) dynamic OT map can be computed from the OT plan (using McCann interpolants), it cannot produce novel trajectories. Indeed, the OT plan is computed between a fixed source and target distributions. Therefore, one can use the OT plan only from the source empirical distribution and it would only generate real target samples. In the paper, we study the problem of generative modelling, where one needs to generate novel samples from an approximation to the data distribution. This has been used in the single-cell setting for example in Cell-OT [Bunne et al. 2023] which trained a neural network from small datasets to approximate the optimal transport pushforward which can be subsequently applied to push forward new source datasets.

We make an analogy with the setting of diffusion models, where given a dataset the true score at any noise level can be evaluated in closed form in O(dataset size) computation. In fact, generating images using this optimal model is not prohibitive for data the scale of CIFAR-10. However, the amortization of this score model into a neural network through stochastic regression induces desirable smoothness properties to which the superior quality of high-dimensional samples generated with continuous-time generative models has been attributed (see, e.g., the analysis in ["Generalization in diffusion models arises from geometry-adaptive harmonic representation", arXiv:2310.02557]).

Similarly, in our setting, we use a stochastic regression to amortize the optimal transport ODE or the probability flow ODE of the Schrödinger bridge -- which can indeed be computed at a given (x,t) given access to source and target distributions and the static OT plan between them -- into a neural network that allows fast generation of new samples. Smoothness is induced both by the neural network approximation and by the minibatch approximation to the static OT.

---

### Decision · Action_Editor_TTMb · 2024-02-06

**Recommendation:** Accept with minor revision

**Comment:**

The paper is well-written and presents nice contributions. My only concern is regarding the comparison with the work of [1].
Indeed, while [1] say that in practice they use independent coupling, they note that any coupling could be used. Therefore, in my opinion, the main generalization of the present paper lies in the use of more general bridges $p_t(x|z)$.

In addition, the comparison with [2] appears to be ambiguous.

As a result, I ask for the authors to clarify their claims in light of this observation and elaborate on their discussion concerning these two papers.

[1] Liu et al 2022 Flow Straight and Fast: Learning to Generate and Transfer Data with Rectified Flow
[2] Michael S. Albergo and Eric Vanden-Eijnden. Building normalizing flows with stochastic interpolants

**Audience:**

Given the current high interest in machine learning for generative models and flow-based methods to relate two given distributions, this paper fit in the scope of TMLR and should find a large audience within the community.

**Claims And Evidence:**

The authors introduce a new class of continuous normalizing flows based that extends and encompasses recent flow-based methods, such as flow matching, rectified flows and stochastic interpolants...
The main objective here is to find an optimal (in the sense of optimal transport) path between two given distributions $p_0$ and $p_1$.

As emphasized by the authors, one limitation of the previous approaches is the initial couplings used for initiate the corresponding algorithms. To circumvent this problem, the authors also suggest to use a mini-batch Sinkhorn strategy to approximate the optimal couplings between $p_0$ and $p_1$.

---

> ### Author Response · Authors · 2024-03-08
> **Camera ready version**
>
> Dear AE,
>
> We are happy to submit the camera ready version of our paper. Following your requests, we have added the following paragraph explaining the difference with [1] and [2]:
>
>
> - `Connection with related Rectified Flow and Stochastic interpolants methods.`
>
> `We note that I-CFM is closely related to the algorithms proposed by \citet{albergo_building_2023,liu_rectified_2022}. In the case where the conditional probability path $p_t$ is a Dirac (\emph{i.e., } $\sigma = 0$), I-CFM is equivalent to \citep{liu_rectified_2022}. Furthermore, if we consider the Gaussian mean $\mu_t = \cos(\frac{1}{2} \pi t) x_0 + \sin (\frac{1}{2} \pi t) x_1$ instead of the linear interpolation, I-CFM would be equivalent to the variance preserving stochastic interpolant in \citet{albergo_building_2023}, which has also been further generalized.`
>
> In addition, we have also added two new competitive methods for our Cifar10 experiments. Namely: Stochastic Interpolants [2] and Variance Preserving Flow Matching [Lipman et al.]. These new experiments confirm that our ICFM and OTCFM methods achieve the best FID score and that OT-CFM has the fastest inference.
>
> We hope that these novel additions will convince you to accept our manuscript.
>
> Best regards,
> The authors